# Absolute Neighbour Difference based Correlation Test for Detecting Heteroscedastic Relationships

**Lifeng Zhang**
School of Information
Renmin University of China
59, Zhongguancun street, Haidian, Beijing, P.R.China 100872
`l.zhang@ruc.edu.cn`

## Abstract

It is a challenge to detect complicated data relationships thoroughly. Here, we propose a new statistical measure, named the absolute neighbour difference based neighbour correlation coefficient, to detect the associations between variables through examining the heteroscedasticity of the unpredictable variation of dependent variables. Different from previous studies, the new method concentrates on measuring nonfunctional relationships rather than functional or mixed associations. Either used alone or in combination with other measures, it enables not only a convenient test of heteroscedasticity, but also measuring functional and nonfunctional relationships separately that obviously leads to a deeper insight into the data associations. The method is concise and easy to implement that does not rely on explicitly estimating the regression residuals or the dependencies between variables so that it is not restrict to any kind of model assumption. The mechanisms of the correlation test are proved in theory and demonstrated with numerical analyses.

## 1 Introduction

Detecting relationships between variables is a fundamental problem of data-driven discovery. The well-known Pearson's correlation coefficient ($Cor$), Spearman's rank correlation, and Kendall's tau are concise, computationally efficient, and theoretically well understood, however can capture only linear or monotonic relationships. Developing more powerful association detection methods has been a challenging research and attracted extensive attention in the past decades.

A number of techniques have been developed to estimate the score of mutual information (MI) as a measure of the dependence between variables, such as bining (partitioning) [19, 20, 11], kernel density estimation (KDE) [17, 25], and k-nearest neighbor distances (kNN) [4, 13]. Distance correlation ($dCor$) is a kind of more general statistics defined on reproducing kernel Hilbert spaces (RKHS) [9, 10]. It has a compact representation analogous to $Cor$, but can capture a wider range of associations [24, 23]. Randomized dependence coefficient (RDC) is a measure of nonlinear dependence between random variables of arbitrary dimension based on the Hirschfeld-Gebelein-Renyi maximum correlation coefficient [16]. Other approaches include maximal correlation [21, 3], kernel canonical correlation analysis (KCCA) [1], principal curve based methods [5, 6], and nonlinear spectral correlation [15]. Recently, [28] introduced an order statistics based association measure called the neighbour correlation coefficient ($nCor$). With sufficient sample size, $nCor$ is able to detect any type of functional relationships and approximate the $R^2$ of the underlying association.

The associations between variables are sometimes too complicated to be quickly and thoroughly identified. Heteroscedasticity is a kind of association pattern that often occurs in real world data and always indicates that an interaction may exist between the independent variable under test and one or more other known or unknown variables. To address this problem, a number of statistical tests have

35th Conference on Neural Information Processing Systems (NeurIPS 2021).

been proposed such as the Park test, the Goldfeld–Quandt test, the Glejser test, the White test, and the Lin-Qu's test in the context of linear, nonlinear, parametric or nonparametric regression analysis [18, 8, 26, 12, 14], and generalized autoregressive conditional heteroscedasticity (GARCH) model for time series data [2, 7].

Nevertheless, these tests are calculated based on the residuals of regression modelling, so that they all rely on explicitly identifying the functional dependency between variables and sometimes can only capture certain types of heteroscedastic residuals depending on the residual models used. All the aforementioned association detection methods are regression free, but fail to diagnose the heteroscedasticity of data explicitly. Even though some approaches such as MI estimation are able to measure the total dependence between variables, they still cannot distinguish heteroscedasticity (nonfunctional relationships) from functional relationships.

Here, we propose a new association measure called absolute neighbour difference based neighbour correlation coefficient ($nCor_{|\Delta|}$) to examine whether the statistical dispersion of a variable is unequal across the range of values of a second variable that predicts it. It can be used together with other association measures to separately detect the impacts of the independent variable on the expect value and variance of the dependent variable, that will obviously lead to a better understanding of the data.

The rest of this study is organized as follows. Sections 2 and 3 briefly review the background of data association and heteroscedasticity, and introduce a new correlation test for examining whether a data relationship is heteroscedastic or not. In Section 4, empirical studies are performed to evaluate the new statistics and make comparisons with previous approaches. Finally, in Section 5, conclusions are drawn to summarize the study. To facilitate the readability of the study, all the relevant theoretical proofs are enclosed in appendices in supplementary material.

## 2 The proposed method

### 2.1 Heteroscedasticity and interaction

Aside from the central tendency, independent variables may sometimes influence the dispersion of dependent variable, which indicates the presence of heteroscedasticity. Consider the general form of a relationship between $(\mathbf{X}, Y)$ expressed as follows.

$$Y = \mathbb{E}[Y|\mathbf{X}] + E = \mathbb{E}[Y|\mathbf{X}] + \sqrt{\mathrm{Var}(E|\mathbf{X})}\Theta = f(\mathbf{X}) + g(\mathbf{X})\Theta \tag{1}$$

where $Y \in \mathbb{R}$, $\mathbf{X} = (X_i|1 \leq i \leq M) \in \mathbb{R}^M$, and $E \in \mathbb{R}$ respectively denote dependent variable, independent variables, and additive noise. $\Theta \in \mathbb{R}$ denotes the normalized noise that is defined as $\Theta = E/\sqrt{\mathrm{Var}(E|\mathbf{X})}$ so that $\mathbb{E}[\Theta] = 0$ and $\mathrm{Var}(\Theta) = 1$. $\mathbb{E}[\cdot|\cdot]$ and $\mathrm{Var}(\cdot|\cdot)$ denote conditional expectation and conditional variance operators respectively. $f(\cdot)$ and $g(\cdot)$ are the two functions respectively characterize how the expectation of $Y$ and the standard deviation of $E$ are expressed conditional on $\mathbf{X}$. For the special case that $E$ is homoscedastic, $\mathrm{Var}(E|\mathbf{X}) = \mathrm{Var}(E)$, and $g(\mathbf{X})$ becomes a constant. Here, we assume that all the $X_i$ are independent to each other, and all the functional relationships either are continuous at all points of their domains or have discontinuities of the first kind.

Heteroscedasticity is a kind of unpredictable dependence which is also referred to as nonfunctional relationship. As defined in (1), it is the nature and manifestation of the interaction of $(\mathbf{X}, \Theta)$ in which the simultaneous influence of the two variables is not additive. In such a situation, $\mathbf{X}$ may distinctly influences the statistical dispersion of $E$ (which equals to $Y$ with its predictable part removed), but nevertheless $E$ cannot be predicted from $\mathbf{X}$ alone by using any function, that is the reason why it is so-called nonfunctional.

Interactions can be diagnosed by using multivariate tests, such as $nCor$, $dCor$, and MI, but only when performed on the exact set of interactive variables. Hence, they are always computational costly to be detected exactly in real applications especially when dealing with large datasets in which there are too many variable subsets that need to be examined one by one. When an interaction occurs, the pairwise relationship between the dependent variable and each interacting independent variable often appear to be heteroscedastic and at least partially nonfunctional. That is to say, a concise heteroscedasticity test may assist in detecting the interactivity of variables which can be described as whether and how much one variable impacts on anther variable through interaction effect. Such a pairwise examination could quickly find out the interacting variables, and does not require any

prior knowledge about the subset partitioning and inter-dependencies of the variables. The detected variables, then, can be used preferentially to perform further multivariate tests for measuring the exact interaction effects. It will obviously reduce the computational cost and make the following detection and analysis more directional and efficient.

## 2.2 The population $nCor_{|\Delta|}$

If a functional relationship exists between $(\mathbf{X}, Y)$, two observations in the domain of $f(\cdot)$ should have similar values of $f(\mathbf{X})$ when the difference between their values of $\mathbf{X}$ are sufficiently small. Suppose there is a set of variables $\mathbf{X}' = (X'_1, \ldots, X'_M)$ that

$$X'_i = X_i + \Delta X_i, \ \forall 1 \leq i \leq M \tag{2}$$

where each $\Delta X_i$ denotes a random variable representing an increment of $X_i$ with extremely small absolute value. Let $\Lambda = \|\Delta \mathbf{X}\| = \|\mathbf{X}' - \mathbf{X}\|$ be the Euclidean norm. It is obviously that $\mathbb{E}[\Lambda]$ can also be made arbitrarily small when each $\mathbb{E}[|\Delta X_i|]$ is sufficiently small, and vice versa. In this sense, $\mathbf{X}'$ can be considered as a near neighbour of $\mathbf{X}$. Correspondingly, the neighbouring dependent variable can be derived as

$$Y' = f(\mathbf{X}') + E' = f(\mathbf{X}) + \Delta f + E' \tag{3}$$

where $E'$ is an independent copy of $E$, that is, $E$ and $E'$ are independent and identically distributed (i.i.d.). $\Delta f$ denotes the difference between the two noise free dependent variables which can be computed as $\Delta f = f(\mathbf{X}') - f(\mathbf{X})$. Then, the neighbour difference $\Delta Y = Y' - Y$ should have the following property.

**Lemma 1.** *Consider a relationship between $(\mathbf{X}, Y)$ as given in* (1). *Let random neighbouring variables $(\mathbf{X}', Y')$ be defined as above. Then, it holds that almost surely*

$$\begin{cases} \lim\limits_{\mathbb{E}[\Lambda] \to 0} \mathbb{E}[\Delta Y] = 0 \\ \lim\limits_{\mathbb{E}[\Lambda] \to 0} \mathrm{Var}(\Delta Y) = 2\mathrm{Var}(E) = 2\mathbb{E}[g(\mathbf{X})^2] \end{cases} \tag{4}$$

Figure 1 shows an example where the heteroscedasticity of one variable arises from the interaction of two other variables. When we investigate the bivariate relationship between $Y$ and either $X_1$ or $X_2$, the other one can be naturally viewed as an unknown noise. It is clearly that when $\Delta X_i$ approaches 0, the expectation of $|\Delta Y|$ implies the conditional dispersion of $Y$ given $X_i$ with $\mathbb{E}[Y|X_i]$ removed. That is to say, the heteroscedasticity of $E$ can be detected by examining whether the expectation of $|\Delta Y|$ is a function of $X_i$ or a constant.

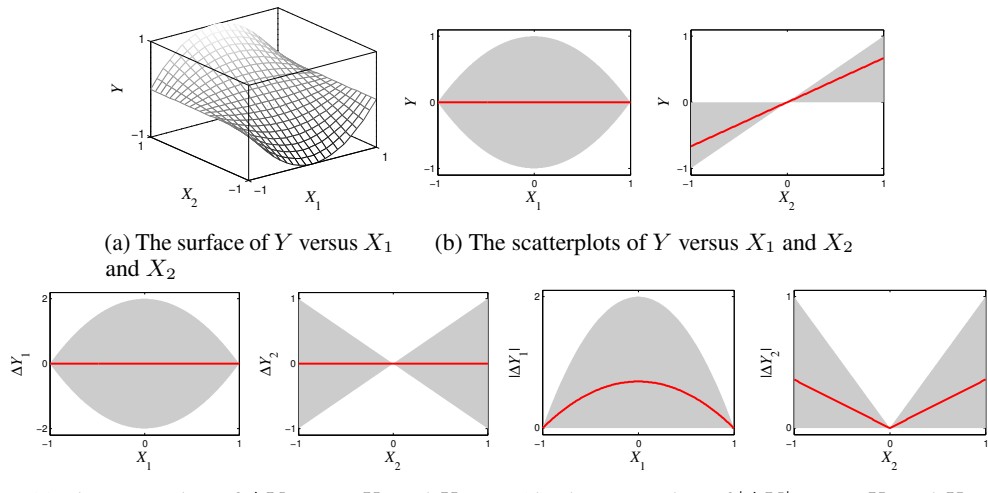

(a) The surface of $Y$ versus $X_1$ and $X_2$    (b) The scatterplots of $Y$ versus $X_1$ and $X_2$

(c) The scatterplots of $\Delta Y$ versus $X_1$ and $X_2$    (d) The scatterplots of $|\Delta Y|$ versus $X_1$ and $X_2$

Figure 1: The surface and scatterplots of $Y = (1 - X_1^2)X_2$, where $X_1, X_2 \in [-1, 1]$. The grey shaded areas and red lines respectively indicate the dispersion of the corresponding variables and their conditional expected values.

Based on this concept, we propose a new correlation test to detect whether a $g(\cdot)$ exists between $(\mathbf{X}, Y - f(\mathbf{X}))$ or not, without modelling either $f(\cdot)$ or $g(\cdot)$ explicitly. Let $\mathbf{X}''$ and $\mathbf{X}'''$ be other two neighbouring variables defined as below.

$$\begin{cases} \mathbf{X}'' = \mathbf{X}' + \Delta\mathbf{X}' \\ \mathbf{X}''' = \mathbf{X}'' + \Delta\mathbf{X}'' \end{cases} \tag{5}$$

where $\Delta\mathbf{X}'$ and $\Delta\mathbf{X}''$ are random increments defined the same as $\Delta\mathbf{X}$. The corresponding dependent variables are defined as

$$\begin{cases} Y'' = f(\mathbf{X}'') + E'' = f(\mathbf{X}') + \Delta f' + E'' \\ Y''' = f(\mathbf{X}''') + E''' = f(\mathbf{X}'') + \Delta f'' + E''' \end{cases} \tag{6}$$

where $E''$ and $E'''$ are independent copies of $E$. Consider the difference $\Delta Y'' = Y''' - Y''$. By Lemma 1, we have that almost surely when $\mathbb{E}[\Lambda''] \to 0$, if $E$ is heteroscedastic, then $\mathrm{Var}(\Delta Y'') \to 2\mathbb{E}[g(\mathbf{X}'')^2]$, otherwise $\mathrm{Var}(\Delta Y'') \to 2\mathrm{Var}(E'')$. Then, by taking advantages of the mechanism of $nCor$, a new association measure is defined as below.

**Definition 1.** *The absolute neighbour difference based neighbour correlation coefficient ($nCor_{|\Delta|}$) when applied to a population is defined as*

$$nCor_{|\Delta|}(\mathbf{X}, Y) = \frac{\mathrm{Cov}(|\Delta Y|, |\Delta Y''|)}{\sqrt{\mathrm{Var}(|\Delta Y|)\mathrm{Var}(|\Delta Y''|)}} \tag{7}$$

**Theorem 1.** *Consider a relationship between $(\mathbf{X}, Y)$ as given in* (1). *Let $(\mathbf{X}', Y')$, $(\mathbf{X}'', Y'')$, and $(\mathbf{X}''', Y''')$ be defined as above. Let $\Lambda$ denotes the vector valued Euclidean norms $\{\|\Delta\mathbf{X}\|, \|\Delta\mathbf{X}'\|, \|\Delta\mathbf{X}''\|\}$. The population $nCor_{|\Delta|}(\mathbf{X}, Y)$ has following properties. First, $|nCor_{|\Delta|}(\mathbf{X}, Y)| \le 1$. Second, if $E$ is homoscedastic, then $nCor_{|\Delta|}(\mathbf{X}, Y) = 0$. Third, if $E$ is heteroscedastic, then almost surely*

$$\lim_{\mathbb{E}[\Lambda] \to \mathbf{0}} nCor_{|\Delta|}(\mathbf{X}, Y) = \frac{\mathrm{Var}\left(g(\mathbf{X})\right)\mathbb{E}\left[|\Theta' - \Theta|\right]^2}{2\mathbb{E}[g(\mathbf{X})^2] - \mathbb{E}\left[g(\mathbf{X})\right]^2\mathbb{E}\left[|\Theta' - \Theta|\right]^2} > 0 \tag{8}$$

See Appendices C and B in supplementary material for the proofs.

**Remark 1.** (8) *clearly suggests that when $\Lambda \to \mathbf{0}$, the value of $nCor_{|\Delta|}$ is determined only by $g(\mathbf{X})$, regardless of whether and what type of $f(\mathbf{X})$ exists. This is contrary to the $nCor$ since $\lim_{\mathbb{E}[\Lambda] \to 0} nCor(\mathbf{X}, Y) = \mathrm{Var}\left(f(\mathbf{X})\right)/\mathrm{Var}(Y) = R^2$ [28], that is, the value of $nCor$ is dependent only on $f(\mathbf{X})$ but completely insensitive to $g(\mathbf{X})$. The two tests have distinct but complementary capabilities, and therefore can be used to respectively examine to what extent $\mathbf{X}$ influences the central tendency and the statistical dispersion of $Y$. In orther words, one is for measuring functional relationship, and the other one is for detecting nonfunctional relationship.*

## 3  The $nCor_{|\Delta|}$ test for a sample

Let $(\mathbf{x}_{(1)}, y_{(1)}), \ldots, (\mathbf{x}_{(N)}, y_{(N)})$ be the paired sequences of $N$ independent observations from the joint distribution of $(\mathbf{X}, Y)$. For computing $nCor_{|\Delta|}$, the realizations of the dummy variables $(\mathbf{X}', Y')$, $(\mathbf{X}'', Y'')$, and $(\mathbf{X}''', Y''')$ can be obtained by resampling from the original dataset, based on the criterion of minimizing the vector valued norms $\lambda$.

### 3.1  Data reordering

In this study, we use the same data reordering methods as used for computing $nCor$ in [28] to generate neighbouring data sequences. The data reordering based resampling process ensures each data point has an equal opportunity to be used in computing the correlation values, and thereby avoids the appearance of a few super data points whose values dominate the resampled sequences due to being copied many times. In this process, each data point is copied four times, respectively, as the observations of the four sets of neighbouring variables. Therefore, the four new samples should obey an identical joint distribution as the original one, which perfectly comply with the concept of neighbouring variables given in Section 3.

In the bivariate case where $M = 1$, the concepts of order statistics and concomitants are presented as follows. First, sorting $\{x_{(t)}\}$ in an increasing order of its values to yield a new sequence denoted by $x_{(1:N)} \leq x_{(2:N)} \leq \cdots \leq x_{(N:N)}$, where $x_{(k:N)}$ is known as the $k$-th order statistic of $\{x_{(t)}\}$. In addition, let $\{n_k | 1 \leq k \leq N\}$ denotes the reordering permutation, that is, if $n_k = t$ then $x_{(k:N)} = x_{(t)}$. Second, rearranging dependent variable data in accordance with $\{n_k\}$ to obtain a new sequence $y_{[1:N]}, y_{[2:N]}, \cdots, y_{[N:N]}$ , where $y_{[k:N]}$ is known as the $k$-th concomitant, and can be defined as $y_{[k:N]} = y_{(t)}$ when $n_k = t$.

For multivariate data ($M > 1$), the optimal reordering permutation can be obtained by minimizing the total distance between each pair of neighbouring data points in $X$ space derived as follows.

$$\{n_k^* | 1 \leq k \leq N\} = \underset{\substack{n_k \in \{1, 2, \cdots, N\}, \forall 1 \leq k \leq N \\ n_i \neq n_j, \forall i \neq j}}{\arg \min} \sum_{k=1}^{N-1} \|\mathbf{x}_{(n_{k+1})} - \mathbf{x}_{(n_k)}\| \tag{9}$$

The nearest neighbor (NN) algorithm, then, can be used to obtain the optimal or most likely a sub-optimal permutation $\{n_k\}$ for generating $\{\mathbf{x}_{(k:N)}\}$ and $\{y_{[k:N]}\}$. The NN algorithm is a kind of constructive search heuristic that is easy to implement and executes quickly. It starts at one data point, then visits the data point that is nearest to the starting point. Afterwards, it visits the nearest unvisited data point, and repeats this process until all data points have been visited. Finally, it outputs the visiting path as the permutation $\{n_k\}$ for data reordering.

## 3.2  Data resampling and correlation test

Let rearranged data sequences $\{(\mathbf{x}_{(k:N)}, y_{[k:N]})\}$, $\{(\mathbf{x}_{(k+1:N)}, y_{[k+1:N]})\}$, $\{(\mathbf{x}_{(k+2:N)}, y_{[k+2:N]})\}$, and $\{(\mathbf{x}_{(k+3:N)}, y_{[k+3:N]})\}$ be considered as the observations of $(\mathbf{X}, Y)$, $(\mathbf{X}', Y')$, $(\mathbf{X}'', Y'')$, and $(\mathbf{X}''', Y''')$ respectively. Then, the neighbour difference can be calculated as $\Delta y_{[k:N-3]} = y_{[k+1:N]} - y_{[k:N]}$ where $1 \leq k \leq N - 3$.

Figure 2 shows four examples that the absolute difference, especially its conditional expectation, provides a more explicit indication of the heteroscedasticity of the data than the direct investigation of the relationship between $(x, y)$.

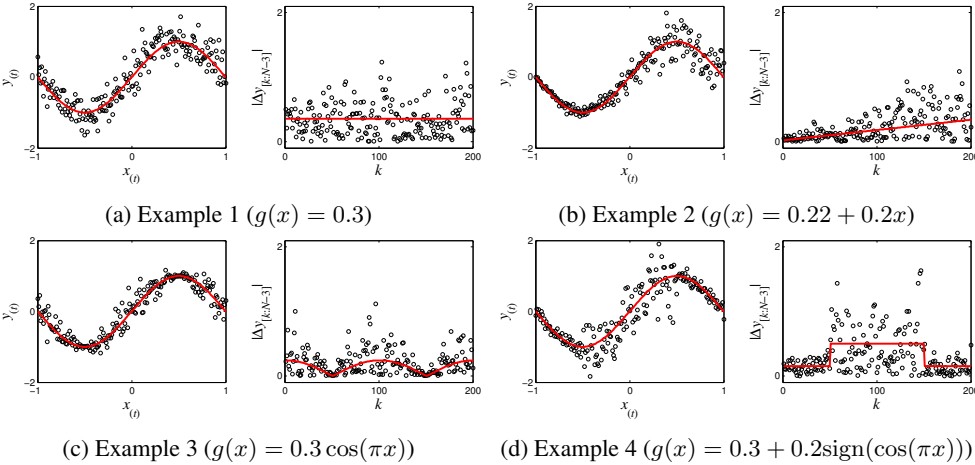

(a) Example 1 ($g(x) = 0.3$)  (b) Example 2 ($g(x) = 0.22 + 0.2x$)

(c) Example 3 ($g(x) = 0.3\cos(\pi x)$)  (d) Example 4 ($g(x) = 0.3 + 0.2\text{sign}(\cos(\pi x))$)

Figure 2: The scatterplots of the dependent variable and absolute difference for four simulated examples with sample size of 200. The examples can be generally represented as $y = \beta \sin(\pi x) + g(x)\theta$, where $x \sim U(-1, 1)$ and $\theta \sim N(0, 1)$. The red lines indicate the conditional Expectations.

**Lemma 2.** *Let $\{(\mathbf{x}_{(t)}, y_{(t)}) | 1 \leq t \leq N\}$ be observed from a relationship as defined in (1), and $\Delta y$ be computed with the optimal reordering permutation as defined in (9). It holds that almost surely,*

$$\begin{cases} \lim_{N \to \infty} \overline{\Delta y} = 0 \\ \lim_{N \to \infty} \text{var}(\Delta y) = 2\text{var}(e) = 2\overline{g(\mathbf{x})^2} \end{cases} \tag{10}$$

*where the overbar and $\text{var}(\cdot)$ respectively denote the mean value and sample variance of the sequence.*

**Definition 2.** *Let $\{A_{[k]}\}$ and $\{B_{[k]}\}$ denote the absolute neighbour difference sequences that is defined as that for any $1 \leq k \leq N-3$*

$$\begin{cases} A_{[k]} = |\Delta y_{[k:N-3]}| = |y_{[k+1:N]} - y_{[k:N]}| \\ B_{[k]} = |\Delta y''_{[k:N-3]}| = |y_{[k+3:N]} - y_{[k+2:N]}| \end{cases} \tag{11}$$

*The $nCor_{|\Delta|}$ when applied to a sample is defined as*

$$nCor_{|\Delta|}(\mathbf{x}, y) = \frac{\sum\limits_{k=1}^{N-3} A_{[k]} B_{[k]} - (N-3)\overline{A}\,\overline{B}}{\sqrt{\sum\limits_{k=1}^{N-3} A_{[k]}^2 - (N-3)\overline{A}^2}\sqrt{\sum\limits_{k=1}^{N-3} B_{[k]}^2 - (N-3)\overline{B}^2}} \tag{12}$$

**Theorem 2.** *Let $\{A_{[k]}\}$ and $\{B_{[k]}\}$ be the absolute neighbour differences computed as given in (11). A hypothesis test rejects the null hypothesis of homoscedastic if*

$$nCor(\mathbf{x}, y) > \tanh\left(\Phi^{-1}(1-\alpha)/\sqrt{(N-6)}\right) \tag{13}$$

*where $\alpha$ denotes the significance level, and $\Phi(\cdot)$ is the standard normal cumulative distribution function. If $e$ is heteroscedastic that a $g(\cdot)$ exists between $x$ and $\mathrm{Var}(e)$, then almost surely*

$$\lim_{N \to \infty} nCor_{|\Delta|}(\mathbf{x}, y) = \frac{\mathrm{var}(g(\mathbf{x}))\overline{|\theta' - \theta|}^2}{2\overline{g(\mathbf{x})^2} - \overline{g(\mathbf{x})}^2\overline{|\theta' - \theta|}^2} > 0 \tag{14}$$

See Appendices C and D in supplementary material for the proofs. In summary, the proposed method is a very concise association measure and only requires three computational steps, that is, reordering the sample points by order statistics or NN algorithm, generating the absolute neighbour difference sequences by (11), and then computing $nCor_{|\Delta|}$ by (12).

**Remark 2.** *(i) $nCor_{|\Delta|}$ is designed to detect the heteroscedasticity of the underlying noise approximated by the absolute neighbour difference, rather than the residuals of any regression model. Hence, it does not require any kind of parametric or nonparametric modelling process. (ii) In most cases, pairwise $nCor_{|\Delta|}$ test may be sufficient for detecting nonfunctional relationships between variables. Multivariate $nCor_{|\Delta|}$ can be used to quickly examine the heteroscedasticity arising from multiple $x_i$ in the context of multivariate regression. When dealling with complex $g(\cdot)$ and high dimensional $\mathbf{x}$, it may exhibit less detection power (as demonstrated in section 4.1). (iii) The proposed test requires comparatively large amounts of samples to ensure the overwhelming majority of data points being close to each other in $\mathbf{x}$ space, otherwise it may display less detection power. In general, several hundreds of data points might be sufficient for conducting the test (as illustrated in section 4.1 and Appendix E in supplementary material).*

**Remark 3.** *When dealing with high-dimensional datasets, finding interacting variables is one of the most important challenges in feature selection and has received wide attention [27, 22]. Interaction effect is a kind of multivariate functional dependence which cannot be directly identified by any pairwise association measure. Since heteroscedasticity always arises from interaction, the bivariate $nCor_{|\Delta|}$ test can also be viewed as a convenient method for detecting interacting variables in the multivariate case. As demonstrated in Appendix G in supplementary material, we can use $nCor_{|\Delta|}(x_i, y)$ to examine the heteroscedasticity of each pair of $(x_i, y)$ for screening the potential interacting $\{x_i\}$, and subsequently conduct the multivariate $nCor$ and the COI tests as introduced in [28] to identify the exact interactions on the reduced independent variable subset that only contains the potential interacting $\{x_i\}$. This procedure will be obviously much more efficient than identifying interactions through detecting the full dataset.*

## 4 Empirical studies

### 4.1 Effectiveness, consistency, and robustness

In this section, we conducted a set of numerical experiments performed on artificial data to evaluate the performance of $nCor_{|\Delta|}$, as well as to make comparisons with previous approaches. The four examples given in Figure 2 were employed as the trial functions. Three data distributions (uniform,

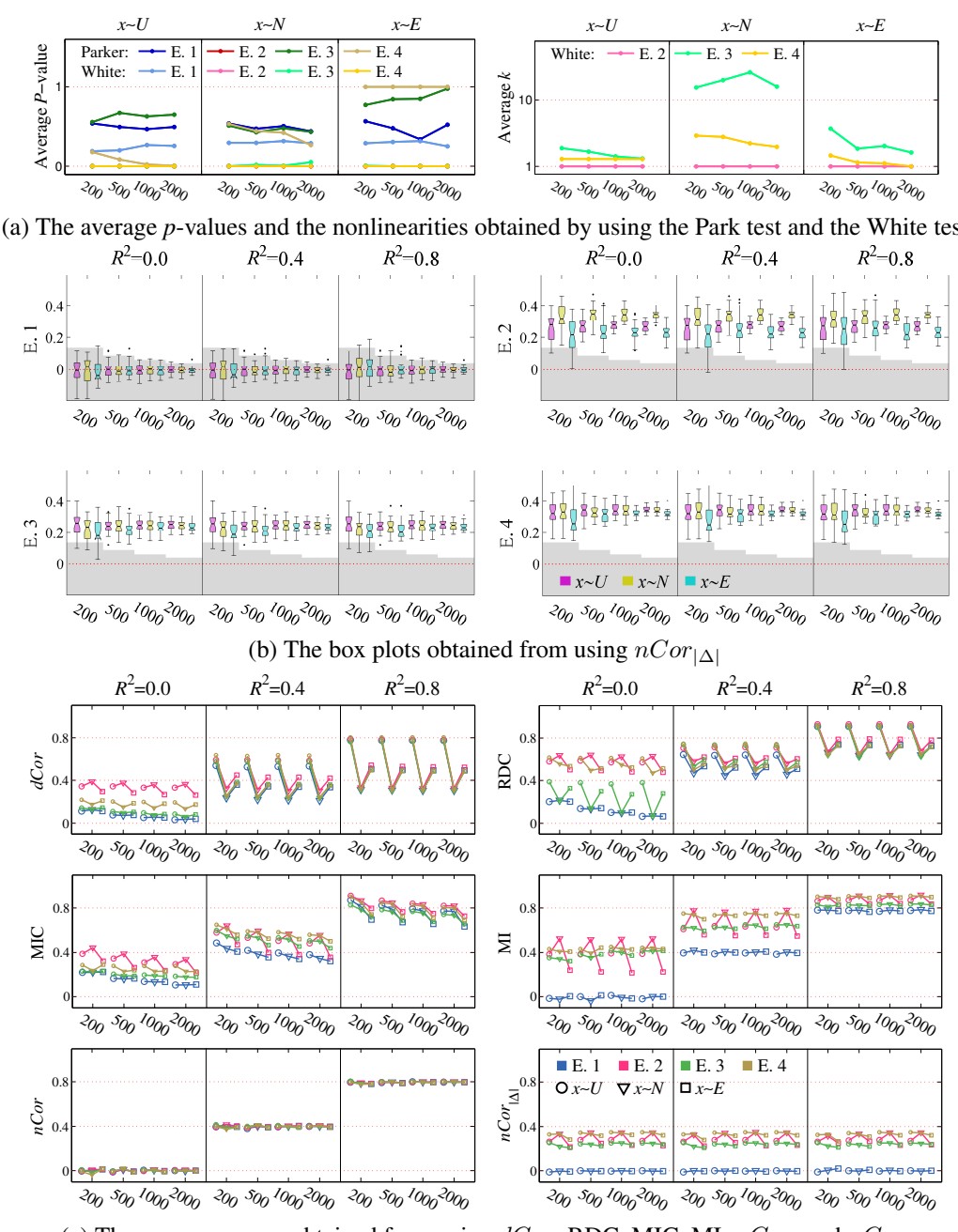

(a) The average *p*-values and the nonlinearities obtained by using the Park test and the White test

(b) The box plots obtained from using $nCor_{|\Delta|}$

(c) The average scores obtained from using $dCor$, RDC, MIC, MI, $nCor$, and $nCor_{|\Delta|}$

Figure 3: The detection results obtained from using the previous heteroscedasticity tests, association measures, and the proposed method for the four examples given in Figure 2 (E. 1 to E. 4). Three data distributions ($U(-1,1)$, $N(0,1)$, and $E(1)$) and four sample sizes (from 200 to 2000) were implemented in these experiments. In (b), the grey shaded areas indicate the 95% confidence intervals.

normal, and exponential), four sample sizes (200, 500, 1000, and 2000), and three $R^2$ (0, 0.4, and 0.8) were considered. For each case, 50 random datasets were generated for conducting the detection.

First, two widely used heteroscedasticity tests, the Park test and the White test (by using polynomials of degree from 1 to 15, and then selecting the lowest *p*-value as the final result), were performed to verify the homogeneity of residuals obtained by assuming the best fitted regression model ($\hat{y}_i = \beta \sin(\pi x)$) was unknown. Figure 3(a) shows that due to the fixed model structure, the Park test failed to reject the hypothesis of homoscedasticity in examples 3 and 4. The nonlinear polynomial based White test successfully detect all the heteroscedastic residuals, however, a rather high degree

of nonlinearity is needed. Especially in example 3, when using normally distributed $x$ only by the polynomials with degree of $k > 10$ can the heteroscedasticity be detected as significant at $p \leq 5\%$. The disadvantage of the two tests and many other approaches of the same kind is that, they all require properly estimated regression residuals and well-specified model assumption so that they are obviously not suitable for quick and convenient association detection.

Second, as shown in Figure 3(b), $nCor_{|\Delta|}$ successfully detected the heteroscedasticity of the data whatever type of underlying $g(x)$ was applied. The vast majority of $nCor_{|\Delta|}$ values are inside the 95% confidence interval in Example 1, and contrarily, the correlation scores are significant in the other three examples. The figure clearly suggests that $nCor_{|\Delta|}$ is a consistent measure that has statistical power increasing as the sample size increases in every case. Moreover, $nCor_{|\Delta|}$ is quite robust to both data distribution and $R^2$, as it exhibited very similar performance with respect to the statistical power under different situations.

To further assess the performance of the proposed method in the extreme situations of small sample size ($N = 30, 50, 100, 150$) and extremely small noise ($R^2 = 0.9, 0.99, 0.999$), more experiments were conducted and the results can be found in Appendix E in supplementary material. The experiment results confirm that $nCor_{|\Delta|}$ may exhibit less detection power under smaller sample sizes, in which case the data points are not guaranteed to be within close distance of each other in $x$. Moreover, the occurrence of outliers (which also lead to large distances between reordered data points) accompanied by extremely large $R^2$ may make the proposed test to be prone to inflated type I error. The experiment results suggest that this problem can be addressed by removing the isolated data points that have extreme values of $x$ from the datasets for $nCor_{|\Delta|}$ computation.

Third, $nCor$ and four typical and widely accepted association measures including $dCor$, RDC, MIC, and kNN based MI estimator were also implemented for comparison purpose. Here, the MI scores were re-scaled to the range of $[0, 1]$ as $1 - exp(-2MI(x, y))$. (i) As shown in Figure 3(c), despite being able to detect $g(x)$ when $R^2 = 0$, $dCor$ and RDC displayed less and less detection power as $R^2$ increases until becoming heteroscedasticity insensitive when $R^2 = 0.8$. (ii) The scores of MIC and MI in the last three examples are perceptibly greater than that in Example 1 even though $R^2 = 0.8$, which should reflect both the dependencies of $f(x)$ and $g(x)$. Despite measuring the total dependence between $(x, y)$, they still cannot distinguish between the impacts of $x$ on the expectation and variance of $y$, in other words, to what extent the dependence is functional (predictable) or not. For example, the scores of MI for E.2 with $R^2 = 0$ are very similar to that for E.1 with $R^2 = 0.4$. Actually, the first dependency is completely unpredictable, and the second one is just the opposite. Both MI and MIC failed to diagnose these conditions. (iii) $nCor$ properly measured the $R^2$ of all the functional relationships, but was completely insensitive to the heteroscedastic relationships. In contrast, $nCor_{|\Delta|}$ showed stable performance on detecting heteroscedasticity whatever $R^2$ is. Clearly, the two tests are ideally complementary to each other, and outperforms the other methods.

Finally, we extended the four examples to multivariate case as given in Figure 4. The figure shows that the detection power of $nCor_{|\Delta|}$ decreases as the dimension of $\mathbf{x}$ increases, and varies with different $g(\cdot)$ and $R^2$ especially when $M$ is large and $f(\cdot)$ is highly nonlinear. This situation can be gradually improved by increasing the sample size, however, it still cannot perform as well as $nCor$ (see Appendix E). This is because, $g(\cdot)$ is more difficult to be detected than $f(\cdot)$ since $g(\cdot)$ is blended with $\theta$ through interaction. In these experiments, moreover, each $x_i$ has exactly the same impact on $y$, which leads to another difficulty that there is no individual heteroscedasticity that dominates the entire relationship. As shown in Appendix E, with large $M$ and nonlinear $g(\cdot)$, the heteroscedastic relationships become too blurred to be diagnosed even by a visual inspection of the scatterplots.

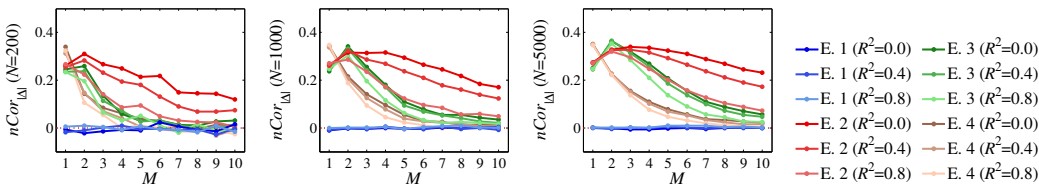

Figure 4: The average $nCor_{|\Delta|}(x_1 \cdots x_M, y)$ scores for $y = \beta \sum_{i=1}^{M} f(x_i) + \sum_{i=1}^{M} g(x_i)\theta$, where $f(\cdot)$ and $g(\cdot)$ were defined the same as in Figure 2, and $x_1$ to $x_M$ were randomly generated with uniform distribution and data length of 200, 1000, 5000.

## 4.2    Detecting complex nonfunctional relationships

In this section, two functional and six nonfunctional associations were employed to further evaluate the proposed method. These data associations were originally introduced by Reshef et al. [19], and most of them were inspired by the patterns found in real-world data, such as line & parabola, ellipse, and non-coexistence. All the data sequences were generated with length of 10000, and corrupted by uniform noises. In the first two experiments, both horizontal and vertical noises with different variances were applied, that is, not only $y$ but also $x$ contain noises. In the last two experiments, noises were separately added into $x$ and $y$. For more details see Appendix F in supplementary material. Figure 5 depicts the simulated data, and the $nCor$ and $nCor_{|\Delta|}$ test results.

For D3-, D4-, D6-, D7-, and D8-, whatever type of noise was applied, the $nCor_{|\Delta|}$ test results suggest that they are heteroscedastic so that there exist strong nonfunctional relationships in these datasets. The $nCor$ test also yielded considerable scores, that is to say, functional relationships coexist so that there is merely a part of the variation of each $y$ that can be predicted from $x$. For D5-, all the $nCor_{|\Delta|}$ scores were significant, and meanwhile, the $nCor$ values felled below the 95% confidence limit. This means that $y$ is completely unpredictable even though an obvious dependence exists between $(x, y)$. In this case, the central tendency of $y$ should be a straight line parallel to $x$ axis. From another perspective, the detection results can also be explained as that in these datasets $x$ influences $y$ through interaction to some extent with other unknown variable(s). As shown in Appendix F, actually, there is another interacting variable $u$ in each case that is invisible but indeed also impacts on $y$. Whenever $u$ is known, the relationship will become fully functional and predictable. In brief, the two tests detected all the relationships in the datasets, and properly revealed the composition of the data associations.

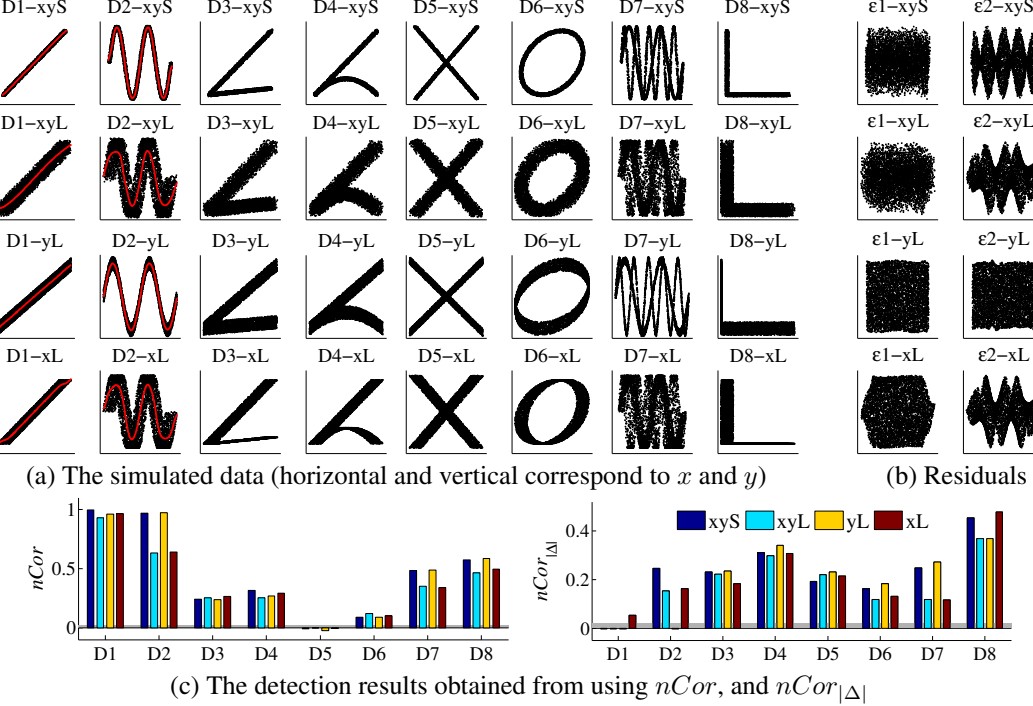

(a) The simulated data (horizontal and vertical correspond to $x$ and $y$)          (b) Residuals

(c) The detection results obtained from using $nCor$, and $nCor_{|\Delta|}$

Figure 5: The scatterplots of the 8 types of bivariate associations corrupted by 4 types of uniform noises. The noise free relationships of D1- and D2- are functional, and the rest are nonfunctional. In D -xyS and D -xyL, both horizontal and vertical noises were added to the data with different noise levels (S and L denotes small and large respectively), by contrast, only vertical noise was added in D -yL, and only horizontal noise was added in D -xL. The red lines in (a) represent the prediction results for the functional relationships obtained by using the best fitted ANN models (3-layers, 3 hidden neurons for D1- and 7 hidden neurons for D2-), and the regression residuals ($\epsilon$) are given in (b). (c) shows the detection results, and the grey shaded areas represent the 95% confidence intervals.

For the functional datasets, D1- and D2-, all the $nCor$ values far exceeded the confidence limit. It is noted that the $nCor_{|\Delta|}$ tests indicate that the underlying noises in four of the eight datasets are

heteroscedastic. It may seem a little counterintuitive that the scatterplots of D2-xyS, D2-xyL, and D2-xL all look more even than that of D2-yL, but nevertheless, by $nCor_{|\Delta|}$ only D2-yL was judged to be homoscedastic and purely functional. This is because in the three datasets the observed $x$ was corrupted by horizontal noise $e_x$, which may lead to an extra interaction term as $y = f(x) + h(x, e_x) + e_y$. Since $e_x$ is unknown, $h(\cdot)$ becomes a heteroscedastic component of the underlying noise and will be remained in regression residuals. Similarly, the test results shows that D1-xL is heteroscedastic, and the heteroscedasticity of D1-xyS and D1-xyL are too weak to be detected due to the corruption by $e_y$. To confirm these findings, feedforward artificial neural network (ANN) was employed to identify the associations and then check on the assumption of homoscedasticity by visual examination of residual plots. Figure 5(a) and 5(b) show the predictive curves and residuals obtained from using ANN models, which are entirely consistent with the results of the $nCor_{|\Delta|}$ test. In brief, $nCor_{|\Delta|}$ successfully prejudged the heteroscedasticity of the residuals without regression modelling.

## 5   Conclusions

Here, we have proposed a new association measure, $nCor_{|\Delta|}$, to detect the heteroscedastic relationships between variables. When used alone, it straightforwardly examines the heteroscedasticity of the underlying noise contained in the data rather than the residuals of a regression analysis, and therefore it is much more efficient and easier to conduct in comparison with the traditional heteroscedasticity tests. When used together with $nCor$, the two tests form an enhanced toolkit that can not only detect a wider range of complex associations, but also distinguish between functional and nonfunctional relationships. Compare to previous association measures, they can obviously get deeper insight into the inter-structure of the data associations, and may provide a promising direction for further analysis. The new method can be widely used to provide solutions for feature selection, model validation, and many other issues in data analysis, interpretation, and modelling.

## Acknowledgement

This work was partially supported by CNPC Northwest Marketing Company (MIS Project No. XBYS-2020-JS-9).

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
