# Supplementary Material of
# Absolute Neighbour Difference based Correlation Test for Detecting Heteroscedastic Relationships

**Lifeng Zhang**
School of Information
Renmin University of China
59, Zhongguancun street, Haidian, Beijing, P.R.China 100872
`l.zhang@ruc.edu.cn`

## Contents

35th Conference on Neural Information Processing Systems (NeurIPS 2021), Sydney, Australia.

# A Proof of Lemma 1

*Proof.* Since $\Delta f$, $E$, and $E'$ are uncorrelated, and $E'$ is a independent copy of $E$ which has zero expectation and finite variance, then the expectation and variance of $\Delta Y$ can be derived as follows.

$$\mathbb{E}[\Delta Y] = \mathbb{E}[Y' - Y] = \mathbb{E}[\Delta f] + \mathbb{E}[E'] - \mathbb{E}[E] = \mathbb{E}[\Delta f] \tag{A1}$$

$$\text{Var}(\Delta Y) = \text{Var}(\Delta f) + \text{Var}(E') + \text{Var}(E) = \text{Var}(\Delta f) + 2\text{Var}(E) \tag{A2}$$

Consider a continuous $f(\cdot)$. Since $\Lambda = \|\Delta \mathbf{X}\|$, then $\Lambda$ is a nonnegative random variable, then we have

$$\begin{aligned}
&\mathbb{E}[\Lambda] \to 0 \Leftrightarrow \Lambda \to 0 \Leftrightarrow \Delta \mathbf{X} \to \mathbf{0}^M \\
&\Rightarrow f(\mathbf{X} + \Delta \mathbf{X}) \to f(\mathbf{X}) \\
&\Leftrightarrow \Delta f \to 0 \\
&\Leftrightarrow \mathbb{E}[\Delta f] \to 0, \text{Var}(\Delta f) \to 0
\end{aligned} \tag{A3}$$

Therefore, it holds almost surely that

$$\begin{cases}
\lim_{\mathbb{E}[\Lambda] \to 0} \mathbb{E}[\Delta Y] = 0 \\
\lim_{\mathbb{E}[\Lambda] \to 0} \text{Var}(\Delta Y) = 2\text{Var}(E)
\end{cases} \tag{A4}$$

If $E$ is heteroscedastic, then $g(X)$ is a variable rather than a constant, and thereby

$$\begin{aligned}
\lim_{\mathbb{E}[\Lambda] \to 0} \text{Var}(\Delta Y) &= 2\mathbb{E}[g(\mathbf{X})^2]\mathbb{E}[\Theta^2] - 2\mathbb{E}[g(\mathbf{X})]^2\mathbb{E}[\Theta]^2 \\
&= 2\mathbb{E}[g(\mathbf{X})^2]\text{Var}(\Theta) = 2\mathbb{E}[g(\mathbf{X})^2]
\end{aligned} \tag{A5}$$

Consider a $f(\cdot)$ that has removable or jump discontinuities. The domain of $f(\cdot)$ can be divided into a set of sub-domains $\mathbf{D} = \{D_i\}$ such that $f(\cdot)$ is continuous on each $D_i$. Then,

$$\mathbb{E}[\Delta f^2] = \sum_{i=1}^{|\mathbf{D}|} \sum_{j=1}^{|\mathbf{D}|} \Pr(\mathbf{X} \in D_i, \mathbf{X}' \in D_j)\mathbb{E}[\Delta f^2 | \mathbf{X} \in D_i, \mathbf{X}' \in D_j] \tag{A6}$$

Since $\lim_{\mathbb{E}[\Lambda] \to 0} \mathbf{X}' = \mathbf{X}$, it holds over the entire domain that

$$\lim_{\mathbb{E}[\Lambda] \to 0} \Pr(\mathbf{X}' \in D_i, \mathbf{X} \in D_j) = \begin{cases} \Pr(\mathbf{X} \in D_i), i = j \\ 0, otherwise \end{cases} \tag{A7}$$

Then,

$$\lim_{\mathbb{E}[\Lambda] \to 0} \mathbb{E}[\Delta f^2] = 0 \times \sum_{i=1}^{|\mathbf{D}|} \Pr(\mathbf{X} \in D_i) + \sum_{i \neq j}^{|\mathbf{D}|} 0 \times \mathbb{E}[\Delta f^2 | \mathbf{X} \in D_i, \mathbf{X}' \in D_j] = 0 \tag{A8}$$

$$\Leftrightarrow \mathbb{E}[\Delta f] = 0, \text{Var}(\Delta f) = 0$$

and thus, (A4) and (A5) still hold.

$\square$

# B Proof of Theorem 1

*Proof.* First, $nCor_{|\Delta|}(\mathbf{X}, Y)$ has the same form as the Pearson product-moment correlation coefficient. According to the Cauchy Schwarz inequality, it should also have a value between $\pm 1$.

Second, consider the numerator of (7).

$$\text{Cov}(|\Delta Y|, |\Delta Y''|)$$
$$= \mathbb{E}[|\Delta Y||\Delta Y''|] - \mathbb{E}[|\Delta Y|]\mathbb{E}[|\Delta Y''|] \tag{A9}$$
$$= \mathbb{E}[|\Delta f + E' - E||\Delta f'' + E''' - E''|] - \mathbb{E}[|\Delta f + E' - E|]\mathbb{E}[|\Delta f'' + E''' - E''|]$$

If $E$ is homoscedastic, then $\Delta f$, $\Delta f''$, $E$, $E'$, $E''$, and $E'''$ are independent with each other, and

$$\mathbb{E}[|\Delta f + E' - E||\Delta f'' + E''' - E''|] = \mathbb{E}[|\Delta f + E' - E|]\mathbb{E}[|\Delta f'' + E''' - E''|]$$
$$\Rightarrow \text{Cov}(|\Delta Y|, |\Delta Y''|) = 0 \tag{A10}$$
$$\Rightarrow nCor_{|\Delta|}(\mathbf{X}, Y) = 0$$

Third, if $E$ is heteroscedastic, it follows from (A9) that

$$\text{Cov}(|\Delta Y|, |\Delta Y''|)$$
$$= \mathbb{E}[|\Delta f + g(\mathbf{X}')\Theta' - g(\mathbf{X})\Theta||\Delta f'' + g(\mathbf{X}''')\Theta''' - g(\mathbf{X}'')\Theta''|] \tag{A11}$$
$$- \mathbb{E}[|\Delta f + g(\mathbf{X}')\Theta' - g(\mathbf{X})\Theta|]\mathbb{E}[|\Delta f'' + g(\mathbf{X}''')\Theta''' - g(\mathbf{X}'')\Theta''|]$$

It can be proved with similar arguments as for Lemma 1 that when $\Lambda \to \mathbf{0}$, it holds that $\Delta f \to 0$, $g(\mathbf{X}') \to g(X)$, $g(\mathbf{X}'') \to g(\mathbf{X})$, and $g(\mathbf{X}''') \to g(\mathbf{X}'')$. Since $g(\mathbf{X}) \geq 0$ and $\mathbb{E}[|\Theta' - \Theta|] = \mathbb{E}[|\Theta''' - \Theta''|]$, then it holds almost surely

$$\lim_{\mathbb{E}[\Lambda] \to 0} \text{Cov}(|\Delta Y|, |\Delta Y''|)$$
$$= \mathbb{E}[g(\mathbf{X})^2]\mathbb{E}[|\Theta' - \Theta|]\mathbb{E}[|\Theta''' - \Theta''|] - \mathbb{E}[g(\mathbf{X})]^2\mathbb{E}[|\Theta' - \Theta|]\mathbb{E}[|\Theta''' - \Theta''|] \tag{A12}$$
$$= \text{Var}(g(\mathbf{X}))\mathbb{E}[|\Theta' - \Theta|]^2 > 0$$

Consider the denominator of (7)

$$(\text{Var}(|\Delta Y|)\text{Var}(|\Delta Y''|))^{0.5} = \left(\mathbb{E}[|\Delta Y|^2] - \mathbb{E}[|\Delta Y|]^2\right)^{0.5} \left(\mathbb{E}[|\Delta Y''|^2] - \mathbb{E}[|\Delta Y''|]^2\right)^{0.5} \tag{A13}$$

When $\Lambda \to 0$, $\Delta Y''$ can be viewed as an independent copy of $\Delta Y$, and therefore

$$\lim_{\mathbb{E}[\Lambda] \to 0} (\text{Var}(|\Delta Y|)\text{Var}(|\Delta Y''|))^{0.5} = 2\mathbb{E}[g(\mathbf{X})^2] - \mathbb{E}[g(\mathbf{X})]^2\mathbb{E}[|\Theta' - \Theta|]^2 \tag{A14}$$

Thus, it holds almost surely that

$$\lim_{\mathbb{E}[\Lambda] \to 0} nCor_{|\Delta|}(\mathbf{X}, Y) = \frac{\text{Var}(g(\mathbf{X}))\,\mathbb{E}[|\Theta' - \Theta|]^2}{2\mathbb{E}[g(\mathbf{X})^2] - \mathbb{E}[g(\mathbf{X})]^2\,\mathbb{E}[|\Theta' - \Theta|]^2} > 0 \tag{A15}$$

$$\square$$

## C Proof of Lemma 2

*Proof.* First, consider the situation where $M = 1$ and $x$ is uniformly distributed on the interval $[a, b]$, therefore, $\Delta x_{(k:N)}$ should obey a beta distribution with the expectation and variance of

$$\begin{cases} \mathbb{E}[\Delta x_{(k:N-1)}] = \dfrac{b - a}{N + 1} \\ \text{Var}(\Delta x_{(k:N-1)}) = \dfrac{N(b - a)^2}{(N + 2)(N + 1)^2} \end{cases} \tag{A16}$$

It follows that almost surely

$$\begin{cases} \lim\limits_{N \to \infty} \mathbb{E}[\Delta x_{(k:N-3)}] = 0 \\ \lim\limits_{N \to \infty} \text{Var}(\Delta x_{(k:N-3)}) = 0 \end{cases} \Rightarrow \lim_{N \to \infty} \Delta x_{(k:N-3)} = 0 \tag{A17}$$

If $f(\cdot)$ is continuous at all points of its domain, by the definition, when $\Delta x_{(k:N-3)} \to 0$, $\Delta f_{[k:N-3]} = f(x_{(k:N)} + \Delta x_{(k:N-3)}) - f(x_{(k:N)}) \to 0$, $\forall 1 \leq k \leq N-3$. By Kolmogorov's strong law of large numbers, $\lim_{N\to\infty} \overline{\Delta y} = \overline{e' - e} = 0$ and $\lim_{N\to\infty} \mathrm{var}(\Delta y) = 2\mathrm{var}(e) = 2\overline{g(x)^2}$ with probability 1.

If $f(\cdot)$ has a finite number of removable or jump discontinuities. The index $k = 1, 2, \ldots, N-2$ can be divided into $m+1$ sub-intervals $[N_1 + 1, N_2], [N_2 + 1, N_3], \ldots, [N_m + 1, N_{m+1}]$ where $N_1 = 0$ and $N_{m+1} = N - 2$, such that $f(\cdot)$ is continuous on each $[x_{(N_i+1:N)}, x_{(N_{i+1}:N)}]$.

$$\overline{\Delta y} = \frac{\sum_{i=1}^{m} \sum_{k=N_i+1}^{N_{i+1}-1} \Delta y_{[k:N]}}{N-3} + \frac{\sum_{i=2}^{m} \Delta y_{[N_i:N]}}{N-3} \tag{A18}$$

Since $m$ and each $\Delta y_{[N_i:N]}$ are finite numbers, then when $N \to \infty$, $\overline{\Delta y} \to 0$. Moreover, it can be proved similarly as above that, when $N \to \infty$, $\mathrm{Var}(\Delta y) \to 2\mathrm{var}(e) = 2\overline{g(x)^2}$ almost surely.

Second, consider the situation where $M > 2$ and each $x_i$ is uniformly distributed on the interval $[a, b]$. Finding the best permutation that satisfies (9) can be viewed as the procedure of solving a travelling salesman problem (TSP), that is, to find the shortest possible path that visits each city (data point) exactly once and returns to the origin one. Consider the Euclidean TSP for $N$ cities uniformly and independently distributed in the $M$-dimensional hypercube of unit volume. Let $L^*(\mathbf{x}_{(1)}, \cdots, \mathbf{x}_{(N)})$ be the length of the shortest path through these cities. The Beardwood-Halton-Hammersley limit law (1959) proved the existence of a universal constant $\beta$ such that with probability 1 [1],

$$\lim_{N\to\infty} \frac{L^*(\mathbf{x}_{(1)}, \cdots, \mathbf{x}_{(N)})}{N^{1-(1/M)}} = \beta \tag{A19}$$

where $0 < \beta < \infty$ is a positive constant that depends only on $M$, and is not known explicitly. For $M = 2$, the best bounds for $\beta$ ($0.625 \leq \beta \leq 0.922$) were originally established by [1] and later improved by some studies [9]. For a larger $M$, $\beta = \sqrt{M/2\pi e}(\pi M)^{1/2M}(1 + O(1/M))$, that is also a finite positive constant [1, 5].

Let $\lambda_{n_k^* n_{k+1}^*} = \|\mathbf{x}_{(n_{k+1})} - \mathbf{x}_{(n_k)}\|$. Since $\{\lambda_{n_k^* n_{k+1}^*}\}$ are i.i.d., then we have

$$\sum_{k=1}^{N} \lambda_{n_k^* n_{k+1}^*} + \lambda_{n_1^* n_N^*} = (b-a)L^*(\mathbf{x}_{(1)}, \cdots, \mathbf{x}_{(N)})$$

$$\Rightarrow \lim_{N\to\infty} \sum_{k=1}^{N} \lambda_{n_k^* n_{k+1}^*} \leq \lim_{N\to\infty} (b-a)\beta N^{1-(1/M)} \tag{A20}$$

$$\Rightarrow \lim_{N\to\infty} \mathbb{E}\left[\lambda_{n_k^* n_{k+1}^*}\right] \leq \lim_{N\to\infty} \frac{(b-a)\beta}{N^{1/M}}$$

$$\Rightarrow \lim_{N\to\infty} \mathbb{E}\left[\lambda_{n_k^* n_{k+1}^*}\right] = 0$$

$\lambda_{n_k^* n_{k+1}^*}$ is a nonnegative random variable, therefore, $\mathbb{E}\left[\lambda_{n_k^* n_{k+1}^*}\right] = 0$ if and only if $\Pr\left(\lambda_{n_k^* n_{k+1}^*} = 0\right) = 1$, that is, $\lim_{N\to\infty} \lambda_{n_k^* n_{k+1}^*} = 0$ almost surely. By the definition of $\lambda_{n_k^* n_{k+1}^*}$, it holds that almost surely,

$$\lim_{N\to\infty} \|\mathbf{x}_{(n_{k+1})} - \mathbf{x}_{(n_k)}\| = 0$$

$$\Rightarrow \lim_{N\to\infty} \Delta\mathbf{x}_{(k:N-3)} = \mathbf{0}^M$$

$$\Rightarrow \lim_{N\to\infty} \Delta y_{[k:N-3]} = e_{[k+1:N-3]} - e_{[k:N-3]} \tag{A21}$$

$$\Rightarrow \lim_{N\to\infty} \overline{\Delta y} = 0, \text{ and } \lim_{N\to\infty} \mathrm{var}(\Delta y) = 2\overline{g(x)^2}$$

Moreovwe, finding the optimal route of an Euclidian TSP instance is in some sense equivalent to finding the optimal path through the points in many small subset of the unit square and then patching these together [1, 5]. Figure A1 shows the graph and the contour map of a functional relationship with $M = 2$.

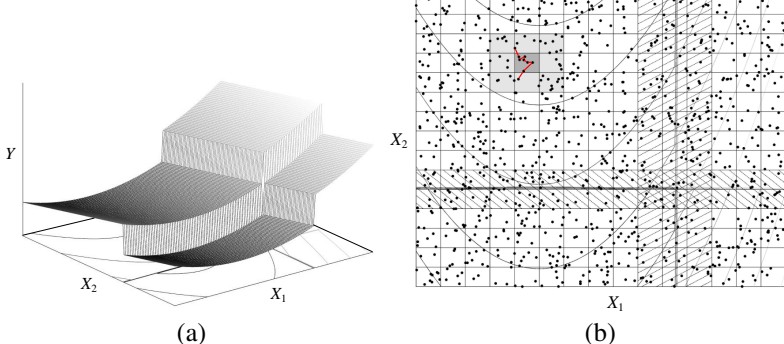

$$(a) \qquad\qquad (b)$$

Figure A1: The surface (a) and contour map (b) of a discontinuous functional relationship between $(x_1, x_2)$ and $y$ which has jump discontinuities at two lines. In figure (b) the grid areas indicate the subdomains surrounding the discontinuities, and the light shaded areas denote the neighbouring subdomains around the dark shaded subdomain. Furthermore, the black dots represent all the data points in a random ample, and the red edges form a short path through the data points within the the dark shaded area .

As shown in the figure, the domain of $f(\cdot)$ (the entire square) can be partitioned into maximum $p \times p$ subdomains (the tiny squares) such that at least one data point falls in almost every subdomain. In this case, each point gets connected only with another point within the same or neighbouring subdomains, so that the edge should have a length of $\lambda_{n_k^* n_{k+1}^*} \leq \sqrt{8}(b-a)/p$. With larger $N$, the entire square can be cut into more tiny squares, which implies that $p$ increases with increasing $N$, while the opposite is true for $\lambda_{n_k^* n_{k+1}^*}$. Therefore, with a sufficiently large $N$, $\lambda_{n_k^* n_{k+1}^*}$ can be limited to an arbitrarily small value.

If $f(\cdot)$ is discontinuous at a finite number of lines in its domain, some of the neighbouring differences may have $\lim_{N \to \infty} \Delta y_{[k:N-3]} \neq e_{[k+1:N-3]} - e_{[k:N-3]}$ due to their corresponding edges being across the discontinuous lines. This kind of edges could occur only between the data points inside the subdomains that lie along the discontinuous lines. Let $P_{discon}$ be the number of the subdomains lying along the lines, and $P_{total}$ be the total number of subdomains. Then, we can easily have $P_{discon} \leq 3lp/(b-a)$ where $l$ is the total length of the discontinuous lines. Since both $l$ and $b-a$ are constant for a given $f(\cdot)$, $P_{discon}/P_{total} \leq 3l/(p(b-a))$ decreases with increasing $p$, and $P_{discon}/P_{total} \to 0$ as $p \to \infty$. When $N \to \infty$, $N_{discon}/N \to 0$, and therefore (A21) still holds. For $M > 2$, it can be proved in the same way as above.

Third, consider the situation where $M = 1$ and $x$ is non-uniformly distributed on the finite interval $[a, b]$. The interval can be divided into a number of sufficiently small subintervals of equal length such that the probability density function of $x$ is monotonic in each subinterval $[\alpha_j, \beta_j]$ as shown in Figure A2. Let $\rho_j$ denotes the maximal value satisfying that $\Pr(\xi_a < x < \xi_b) \geq \rho_j(\xi_b - \xi_a)$ for any $\xi_a < \xi_b$ and $\xi_a, \xi_b \in [\alpha, \beta_j]$. Consider a variable $v$ that is uniformly distributed on $[v_a, v_b]$ where $v_a \leq \alpha_j$, $v_b \geq \beta_j$, and $v_b - v_a = 1/\rho_j$. For any $x_{(k:N-3)} \in [\alpha, \beta_j]$, if there exists a $v_{(q:N-3)} = x_{(k:N-3)}$, then it should have $\Pr(v_{(q:N-3)} < x < v_{(q:N-3)} + \delta) \geq \Pr(v_{(q:N-3)} < v < v_{(q:N-3)} + \delta)$ for any $0 < \delta \leq \beta_j - v_{(q:N-3)}$, that is to say, $\mathbb{E}[\Delta x_{(k:N-3)}] \leq \mathbb{E}[\Delta v_{(q:N-3)}]$. By (A16) and (A17), $\lim_{N \to \infty} \Delta v_{(q:N-3)} = 0$, then $\lim_{N \to \infty} \Delta x_{(k:N-3)} = 0$, and thereby $\lim_{N \to \infty} \overline{\Delta y} = \overline{e' - e} = 0$, and $\lim_{N \to \infty} \text{var}(\Delta y) = 2\text{var}(e) = 2\overline{g(x)^2}$ almost surely.

For $M > 1$, it can be proved similarly as above by dividing the space of $\mathbf{x}$ into a number of subregions. In brief, when $N$ is sufficiently large, the sample points can be made any crowded in $\mathbf{x}$, and the distance between each pair of neighbouring points space can be reduced to arbitrary short.

$$\square$$

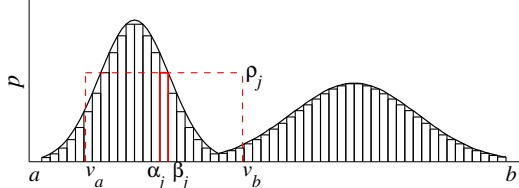

Figure A2: Sketch plot of dividing the interval of $x$ into a number of subintervals

## D    Proof of Theorem 2

*Proof.* If a noise is homoscedastic, as proved in Theorem 1, the expectation of $nCor_{|\Delta|}(\mathbf{x}, y)$ is zero. $nCor_{|\Delta|}$ is in the form of the Pearson correlation coefficient, then, it should also have the same basic properties such as $|nCor_{|\Delta|}| \leq 1$.

By using the Fisher transformation, the confidence limits of correlation estimates with sample size of $N - 3$ are $\pm \tanh\left(\Phi^{-1}(1 - \alpha/2)/\sqrt{(N - 6)}\right)$. A test rejects the independence of $\{(A_{[k]}, B_{[k]})\}$ with a significance level of $\alpha$ when $|nCor_{|\Delta|}(\mathbf{x}, y)| > \tanh\left(\Phi^{-1}(1 - \alpha/2)/\sqrt{(N - 6)}\right)$.

Consider a heteroscedastic noise. If both $f(\cdot)$ and $g(\cdot)$ are continuous, it can be proved similarly as in Lemma 2 that when $N \to \infty$, for all $1 \leq k \leq N - 3$, $A_{[k]} \to g_{[k:N]}|\theta_{[k+1:N-3]} - \theta_{[k:N-3]}|$ and $B_{[k]} \to g_{[k:N]}|\theta_{[k+3:N-3]} - \theta_{[k+2:N-3]}|$. Moreover, $\overline{|\theta' - \theta|} = \overline{|\theta''' - \theta''|} + \left(|\theta_{[2:N]} - \theta_{[1:N]}| + |\theta_{[3:N]} - \theta_{[2:N]}| - |\theta_{[N-3:N]} - \theta_{[N-4:N]}| - |\theta_{[N-2:N]} - \theta_{[N-3:N]}|\right)/(N - 3)$ , hence, $\lim_{N \to \infty} \overline{|\theta''' - \theta''|} = \overline{|\theta' - \theta|}$.

Then, the limit of the numerator of (12) can be derived as

$$\lim_{N \to \infty} \sum_{k=1}^{N-3} A_{[k]} B_{[k]} - (N - 3)\overline{A}\,\overline{B} = (N - 3)\left(\mathrm{var}(g(\mathbf{x}))\overline{|\theta' - \theta|}^2 + c\right) \tag{A22}$$

where

$$\begin{aligned}
c = &\overline{g(\mathbf{x})^2}\mathrm{cov}(|\theta' - \theta|, |\theta''' - \theta''|) \\
&+ \mathrm{cov}(g(\mathbf{x})^2, |\theta' - \theta||\theta''' - \theta''|) \\
&+ \overline{g(\mathbf{x})}\,\overline{|\theta' - \theta|}\,\mathrm{cov}(g(\mathbf{x}), |\theta''' - \theta''|) \\
&+ \overline{g(\mathbf{x})}\,\overline{|\theta''' - \theta''|}\,\mathrm{cov}(g(\mathbf{x}), |\theta' - \theta|) \\
&+ \mathrm{cov}(g(\mathbf{x}), |\theta' - \theta|)\,\mathrm{cov}(g(\mathbf{x}), |\theta''' - \theta''|)
\end{aligned} \tag{A23}$$

Since $\theta$ is a random noise and independent to $g(\mathbf{x})$, all the covariance terms in (A23) should have zero expectation. By Kolmogorov's strong law of large numbers, $\lim_{n \to \infty} c = 0$ with probability 1. since $\lim_{N \to \infty} \sum_{k=1}^{M} A_{[k]}^2 = \lim_{N \to \infty} \sum_{k=1}^{M} B_{[k]}^2 = 2(N - 3)\overline{g(\mathbf{x})^2}\mathrm{var}(\theta)$ and $\lim_{N \to \infty} \overline{A}^2 = \lim_{N \to \infty} \overline{B}^2 = \overline{g(\mathbf{x})}^2\overline{|\theta' - \theta|}^2$, then it holds almost surely

$$\lim_{N \to \infty} \sum_{k=1}^{N-3} A_{[k]} B_{[k]} - (N - 3)\overline{A}\,\overline{B} = (N - 3)\mathrm{var}(g(\mathbf{x}))\overline{|\theta' - \theta|}^2$$

$$\Rightarrow \lim_{N \to \infty} nCor_{|\Delta|}(\mathbf{x}, y) = \frac{\mathrm{var}(g(\mathbf{x}))\overline{|\theta' - \theta|}^2}{2\overline{g(\mathbf{x})^2} - \overline{g(\mathbf{x})}^2\overline{|\theta' - \theta|}^2} > 0 \tag{A24}$$

If $f(\cdot)$ and $g(\cdot)$ have removable or jump discontinuities (or $\mathbf{x}$ are distributed non-uniformly), the domain of $f(\cdot)$ and $g(\cdot)$ can be divided into a number of subdomains such that both $f(\cdot)$ and $g(\cdot)$ are continuous (or $\mathbf{x}$ approximately obey a multivariate uniform distribution) in each subdomain. In such a situation, by Lemma 2, when $N \to \infty$, $\Delta y$ reamins zero mean and variance of $2\overline{g(x)^2}$ (which implies that $\Delta y$ is just a superposition of two realizations of $e$), and then it can be proved similarly as above that (A24) still holds.

□

# E   Supplementary to Section 4.1

**The first experiment:** we applied two widely used heteroscedasticity tests, the Park test and the White test to detect the relationships between $x$ and $\epsilon = e$ (the underlying noise, representing the residuals obtained by assuming $f(\cdot)$ is known). The White test is realized by using polynomials expressed as below. Due to the high nonlinearity of $g(x)$ in example 3, the maximum degree of the White test was set to be 15, otherwise it may fail to detect the heteroscedasticity of the residuals.

$$\hat{\epsilon} = a_0 + a_1 x^1 + \cdots + a_k x^k \tag{A25}$$

**The second experiment:** we considered four $R^2$ values (0, 0.2, 0.4, 0.8) to evaluate the proposed method under different strength of functional relationship coexisted in the data. Here, $R^2$ was directly measured as $Var(\beta \sin(\pi x))/Var(y)$, and meanwhile different levels of $R^2$ were exactly achieved by adjusting the value of $\beta$ for each dataset.

We also evaluated the performance of the proposed test under smaller sample sizes and extremely large $R^2$, and the experiment results are given in Figure A3. As shown in the figure, $nCor_{|\Delta|}$ showed less detection power under smaller sample sizes, in which case the distances in $x$ between the neighbouring data points were not guaranteed to be sufficiently close. Moreover, when $R^2$ was extremely large and $x$ was nonuniformly distributed, the $nCor_{|\Delta|}$ incorrectly rejected the null hypothesis of homoscedastic in E.1. This is because when 99% or even 99.9% of the variance of $y$ is predictable from $x$ by $f(x)$ and thus $e$ becomes extremely small, $\Delta f$ between the outlier data points (which have extreme values of $x$ and thereby large $\Delta x$) may considerably impact the estimation of $nCor_{|\Delta|}$. To address this problem, we removed 10% data points which were of the $x$ values that were the most far away from the mean value of $x$, and then used the reduced datasets to implement the proposed test. The figure clearly suggests that by removing the isolated data points the performance of $nCor_{|\Delta|}$ became acceptable especially with larger sample size.

**The Third experiment:** Four existing association measures were also implemented for make comparisons with the proposed method. These approaches are typical and well-established. MI is the most well-known dependence measure developed in the context of information theory. $dCor$ can be viewed as a special case of kernel based method [8], which is a kind of more general statistic defined on reproducing kernel Hilbert spaces [2, 3]. The basic idea behind MIC is that if an association exists between two variables, then a grid can be drawn on the scatterplot that partitions the data to encapsulate that relationship. Its outstanding performance has been extensively evaluated [6, 7]. RDC enables an effective measure of non-linear dependence between random variables of arbitrary dimension based on the Hirschfeld-Gebelein-Rényi Maximum Correlation Coefficient[4]. The source code in Matlab for $dCor$ and $k$-NN based MI estimator are enclosed in this supplementary material. The R code for conduting RDC can be found at https://github.com/lopezpaz/randomized/dependence/coefficient. The MINE toolbox for implementing MIC test can be found at http://www.exploredata.net/Downloads/MINE-Application. The number of nearest neighbors $k$ of the kNN based MI was set to be 1. The parameters of RDC were set as $k = 20$ and $s = 1/6$.

**The last experiment:** we evaluate the performance of the proposed method under multiple independent variables. The four underlying functions are give as follows.

Table A1: The four underlying functions used as E.1 to E.4.

$$y = \beta \sum_{i=1}^{M} \sin(\pi x) + 0.3\theta$$
$$y = \beta \sum_{i=1}^{M} \sin(\pi x) + \sum_{i=1}^{M} (0.22 + 0.2x)\theta$$
$$y = \beta \sum_{i=1}^{M} \sin(\pi x) + 0.3 \sum_{i=1}^{M} \cos(\pi x)\theta$$
$$y = \beta \sum_{i=1}^{M} \sin(\pi x) + \sum_{i=1}^{M} (0.3 + 0.2\text{sign}(\cos(\pi x)))\,\theta$$

Figure A4 shows the detection results obtained from using the $nCor_{|\Delta|}$ and $nCor$ tests. Figure A5 depicts scatterplots of the observed data describing the relationships of $(x_i, y)$ and $(x_i, e)$. The figure clearly suggests that the $nCor$ lines of the same $R^2$ are virtually superimposable in every case although with different $g(\cdot)$. In contrast, the values of the $nCor_{|\Delta|}$ mainly depend on whether or what type of $g(\cdot)$ exists. Both the two association measures displayed a decreasing detection power as the dimension of $\mathbf{x}$ increased. Actually, all the multivariate association measures has this

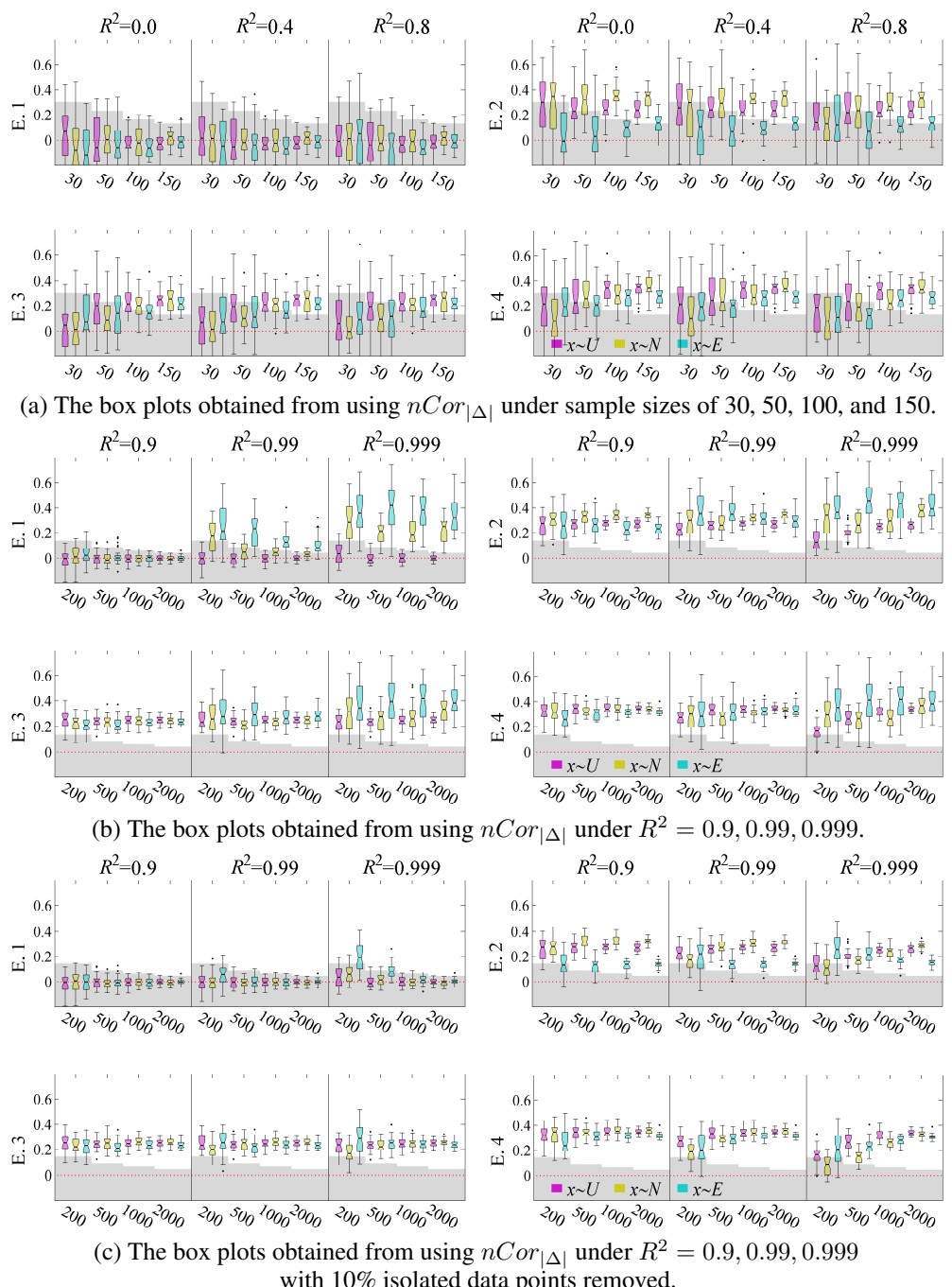

(a) The box plots obtained from using $nCor_{|\Delta|}$ under sample sizes of 30, 50, 100, and 150.

(b) The box plots obtained from using $nCor_{|\Delta|}$ under $R^2 = 0.9, 0.99, 0.999$.

(c) The box plots obtained from using $nCor_{|\Delta|}$ under $R^2 = 0.9, 0.99, 0.999$ with 10% isolated data points removed.

Figure A3: The detection results obtained from the proposed method for the four examples given in Figure 2 (E. 1 to E. 4) under smaller sample sizes and extremely large $R^2$.

common problem. With a higher dimension, the data points in $\mathbf{x}$ space is more sparse and scattered such that the relationship becomes indistinct and difficult to detect. Compared to the $nCor$, the $nCor_{|\Delta|}$ exhibited less detection power on multivariate data especially when $g(\cdot)$ is nonlinear. This is because the heteroscedasticity of $e$ arises from the interaction with an unknown noise $\theta$ so that the heteroscedastic relationships become too blurred to be diagnosed when $M$ is large and $g(\cdot)$ is highly nonlinear. For E.4 with $M = 6$, as shown in Figure A5, the heteroscedasticity of the noise is barely perceptible by visual inspection of the scatterplot.

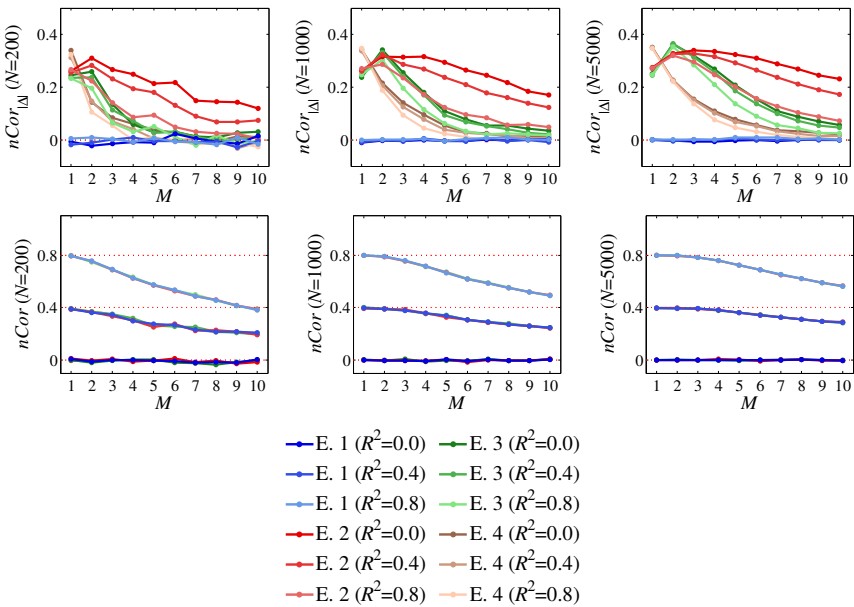

Figure A4: Detection results obtained from using $nCor_{|\Delta|}$ and $nCor$

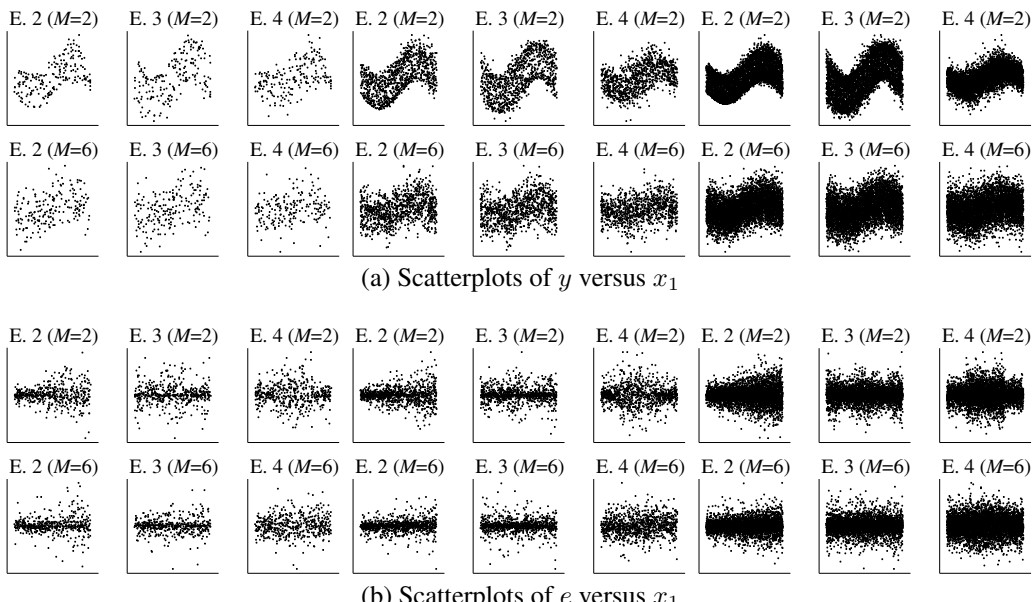

(a) Scatterplots of $y$ versus $x_1$

(b) Scatterplots of $e$ versus $x_1$

Figure A5: The sccaterplots of the observed data with sample sizes of 200, 1000, and 5000.

# F Supplementary to Section 4.2

Table A2 represents the eight associations adopted in Subsection 4.2 for generating simulation data. $x'$ in D 1 to D 8 except D 6, and $y'$ in D 8 were set to be uniformly distributed rand variables with range of $[0\ 1]$. Additive noise $e_x$ and $e_y$ were also uniformly distributed with zero mean and finite variance. In D 7, $t$ was uniformly distributed with range of $[0\ 2\pi]$. In other cases, $u$ was set to be a uniformly distributed binary variable. Moreover, when using ANN models to approximate the underlying associations of D1 and D2, to avoid the problem of overfitting, original data was randomly divided into a training set (70%) and a validation set(30%) for performing earlier stopping strategy. In each case, 10 ANNs were trained and the ANN model with the lowest root-mean-squared-error

(RMSE) was selected as the best fitted regression model. Finally, ANN modelling was conducted using Matlab deep learning toolbox.

Table A2: Eight sets of functions for generating simulation datasets used in Subsection 4.2.

$$\text{D 1} \quad \begin{cases} y = x' + e_y \\ x = x' + e_x \end{cases}$$

$$\text{D 2} \quad \begin{cases} y = sin(4\pi x') + e_y \\ x = x' + e_x \end{cases}$$

$$\text{D 3} \quad \begin{cases} y = \begin{cases} x' + e_y, & u = 0 \\ 0.1x' + e_y, & u = 1 \end{cases} \\ x = x' + e_x \end{cases}$$

$$\text{D 4} \quad \begin{cases} y = \begin{cases} x' + e_y, & u = 0 \\ 0.25 - (x' - 0.5)^2 + e_y, & u = 1 \end{cases} \\ x = x' + e_x \end{cases}$$

$$\text{D 5} \quad \begin{cases} y = \begin{cases} 2(x' - 0.5) + e_y, & u = 0 \\ -2(x' - 0.5) + e_y, & u = 1 \end{cases} \\ x = x' + e_x \end{cases}$$

$$\text{D 6} \quad \begin{cases} u = \sin(t) \\ x' = \cos(t) \\ y = 5.4 + 3u + 0.6x' + e_y \\ x = x' + e_x \end{cases}$$

$$\text{D 7} \quad \begin{cases} y = \begin{cases} sin(4\pi x'), & u = 0 \\ sin(10\pi x'), & u = 1 \end{cases} \\ x = x' + e_x \end{cases}$$

$$\text{D 8} \quad \begin{cases} y = uy' + e_y \\ x = (1 - u)x' + e_x \end{cases}$$

## G    Detecting heteroscedastic relationships in real-world datasets

We used the $nCor$ and $nCor_{|\Delta|}$ tests to explore a real-world large dataset which consists of 356 social, economic, health, and political indicators (variables) of 202 countries (samples) collected from the World Health Organization (WHO) and partner organizations [1,2]. We detected tens of millions of bivariate (126,380) and trivariate (22,369,260) relationships, and found a huge number of heteroscedastic relationships (9,739) and many interesting associations including interaction effects. Subsequently, ANN models was used to identify the relationships to confirm the detection results. In these experiments, three layer feedforward ANN structure was adopted, and the hidden neuron number was set to be 3 and 5 respectively for identifying bivariate and trivariate relationships.

Four examples of the detected associations are given in Tables A3 and A4 and Figure A6. Figure A6 clearly shows that in all the examples, $nCor_{|\Delta|}$ successfully detected the heteroscedasticity of the underlying noises (residuals), which were subsequently verified by ANN base regression. Second, $nCor$ roughly approximate the $R^2 = \text{var}(\hat{f}(\cdot))/\text{var}(y)$ of the functional relationships. By using the two measures together we can easily distinguish whether $x_a$ or $x_b$ influences the expected value or variance of $x_c$. Third, as given in Table A4, when the heteroscedastic noises occurred, there might exist interaction effects between the variables.

## H    Source code

The source code in Matlab for conducting the $nCor$ and $nCor_{|\Delta|}$ tests are also given as below.

**Source code:**

```
%————————————————————————————
function [r1,r2] = ncor2(x,y)
% ncor2 returns the nCor and nCor_{|Δ|} scores between x and y
```

Table A3: The 12 variables that were used in the four examples in Figure A.5

| Notation | Variable name |
|---|---|
| x-1 | Continent |
| x-309 | Old version of income per person |
| x-107 | Age-standardized mortality rate for cardiovascular diseases (per 100000 population) |
| x-11 | Population median age (years) |
| x-336 | Stomach cancer deaths per 100000 men |
| x-58 | Per capit total expenditure on health at average exchange rate (USD) |
| x-9 | Population in urban areas |
| x-227 | Female labour force |
| x-276 | Lung cancer new cases per 100000 women |
| x-108 | Age-standardized mortality rate for injuries (per 100000 population) |
| x-121 | Healthy life expectancy (HALE) at birth (years) both sexes |
| x-140 | Years of life lost to injuries |

Table A4: Association detection and regression modelling results for the four examples. $\Delta R^2$ denotes the increment of $R^2$ when using the two independent simultaneously, which was computed as $\Delta R^2 = R^2_{x_a, x_b \rightarrow x_c} - R^2_{x_a \rightarrow x_c} - R^2_{x_b \rightarrow x_c}$. $\Delta R^2$ equals to the total $R^2$ of $\hat{f}_{a,b}(x_a, x_b)$ with the individual $R^2$ of the two main effects of $\hat{f}_a(x_a)$ and $\hat{f}_b(x_b)$ removed so that it directly implies the strength of the interaction effect of $x_a$ and $x_b$

| Relationships | $nCor_{|\Delta|}$ | $nCor$ | $R^2$ | $\Delta R^2$ |
|---|---|---|---|---|
| (x-1,x-107) | 0.310 | 0.168 | 0.133 | – |
| (x-309,x-107) | 0.300 | 0.232 | 0.386 | – |
| (x-1,x-309,x-107) | – | 0.684 | 0.667 | 0.148 |
| (x-11,x-58) | 0.300 | 0.435 | 0.434 | – |
| (x-336,x-58) | 0.208 | 0.084 | 0.147 | – |
| (x-11,x-336,x-58) | – | 0.755 | 0.726 | 0.145 |
| (x-9,x-276) | 0.171 | 0.116 | 0.278 | – |
| (x-227,x-276) | 0.290 | 0.096 | 0.053 | – |
| (x-9,x-227,x-276) | – | 0.559 | 0.527 | 0.196 |
| (x-108,x-140) | 0.325 | 0.013 | 0.058 | – |
| (x-121,x-140) | 0.288 | 0.318 | 0.387 | – |
| (x-108,x-121,x-140) | – | 0.715 | 0.762 | 0.316 |

```
%    x has N rows and M columns and y has N rows and one column
%    r1 and r2 respectively denotes the nCor and nCor|Δ| scores

% Check the sizes of x and y
if size(x,1)  = size(y,1)
    error('x and y must have the same number of rows')
elseif size(y,2)>1
    error('y must have one column')
end

% Normalize x before computing distance matrix
if size(x,2)>1
    for i = 1:size(x,2)
        x(:,i) = x(:,i)-mean(x(:,i));
        x(:,i) = x(:,i)/std(x(:,i));
    end
end

% Generate permutation by order statistics or NN algorithm
if size(x,2)>1
    dis = squareform(pdist(x));
    perm = computePerm(dis,1);
```

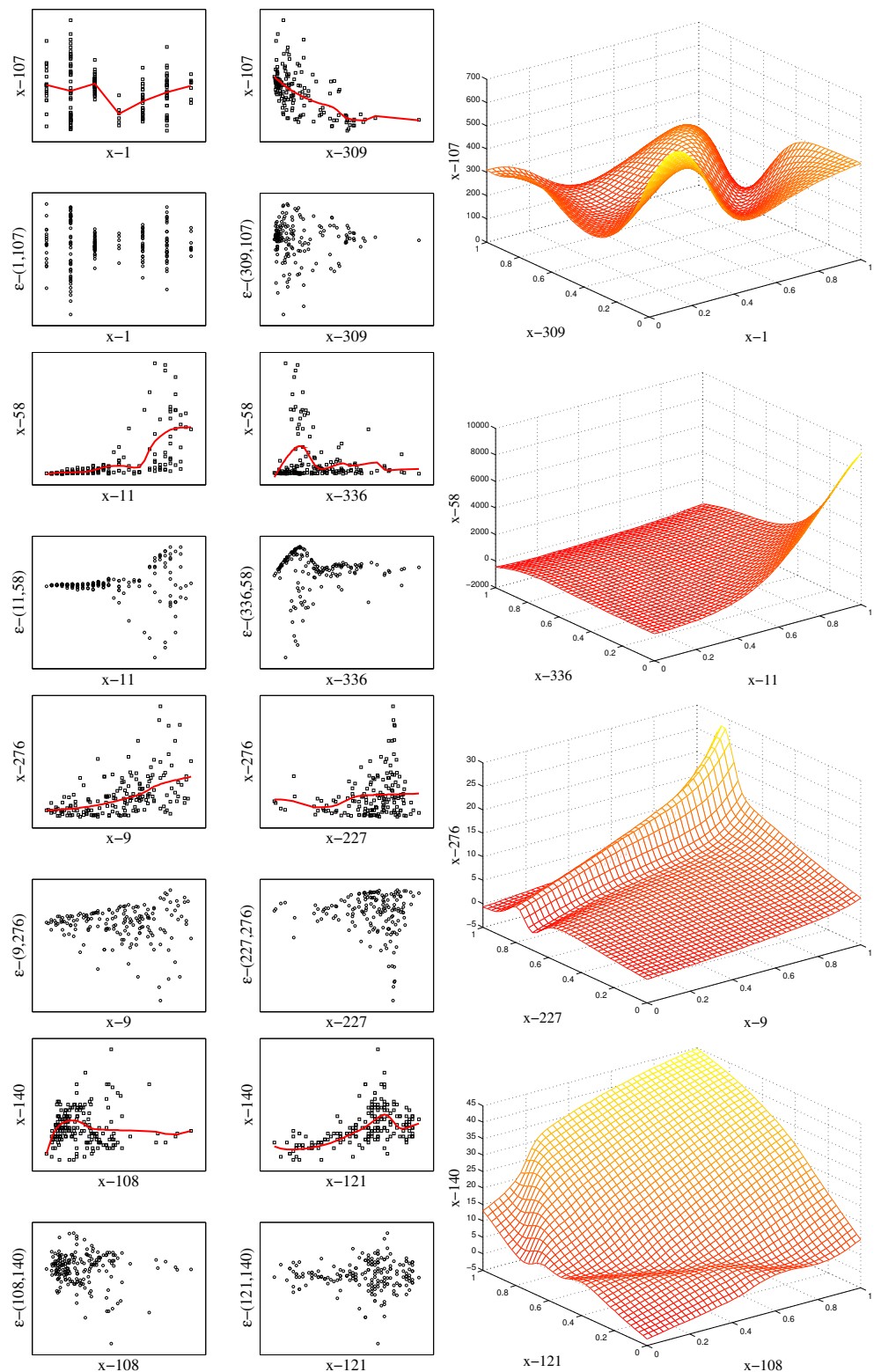

Figure A6: The scatterplots of the four examples. The red lines and surfaces respectively represent the predicted bivariate and trivariate relationships obtained from using the best fitted ANNs.

```
else
    [∼,perm] = sort(x);
end

% Compute the correlation values
y = y(perm);
temp = corrcoef(y(1:end-1),y(2:end));
r1 = temp(1,2); % the nCor score
y1 = abs(y(2:end-2)-y(1:end-3));
y2 = abs(y(4:end)-y(3:end-1));
temp = corrcoef(y1,y2);
r2 = temp(1,2); % the nCor_|Δ| score

%————————————————————————
function perm = computePerm(dis)
% Generate permutation using NN algorithm
perm = 1+zeros(1,length(dis));
N = 2:length(dis);
for i = 2:length(perm)
    [∼,temp] = min(dis(perm(i-1),N));
    perm(i) = N(temp);
    N(temp) = [];
end
```