# OpenReview forum: "Absolute Neighbour Difference based Correlation Test for Detecting Heteroscedastic Relationships"
_NeurIPS.cc/2021/Conference — NeurIPS 2021 Poster_

### Official Review · Reviewer_Xjiq · 2021-07-16

**Rating:** 7
**Confidence:** 3

**Summary:**

This paper proposed a new statistical measure, called absolute neighbour difference based correlation test, to detect the existence of heteroscedastic noise in the residual of a functional relationship.

Theorem 1 shows the proposed measure nCor_Delta is only affected by the dispersion function of the residual g(x) and not affected by the functional relationship f(x). As a result, compared to those existing statistical test of heteroscedastic noise, the major advantage of the proposed measured is to avoid fitting the regression functions. The estimator of the measure on real-world datasets is also presented in Definition 2. The statistical test of heteroscedastic noise vs. homoscedastic noise is presented in Theorem 2.

In the experimental results, the proposed nCor_Delta is firstly compared against two existing tests of heteroscedastic noise, including Parker test and White test. It shows the nCor_Delta can correctly test for heteroscedastic noise while its competing alternatives either fail on some datasets or uses over-complicated regression functions.

The proposed nCor_Delta is also evaluated against a few existing non-linear dependency measures that are not designed to test heteroscedastic noise, including dCor, RDS, MIC, MI and nCor. These measures cannot detect heteroscedastic noise as expected.

**Ethical Concerns:**

No.

**Ethics Review Area:**

["I don’t know"]

**Limitations And Societal Impact:**

The author did clarify the limitations of this work in terms of less power in high-dimensional datasets. Since this is a general statistical test, I didn't see any negative societal impact of this work.

**Main Review:**

The key contribution of this work, i.e. a new statistical test for heteroscedastic noise, seems novel. The writings of the major contributions of related works are also quite clear.

Could the author elaborate on how to combine nCor_Delta and nCor to better discover relationships between variables? For example, if nCor_Delta can detect the heteroscedastic noise between input X and output Y. Given this discovery, how to apply nCor to find a better functional relationship compared to directly applying nCor?

In section 4.1, 50 datasets are created for each functional relationship. However, in real-world data, only a single realization is available. Can the authors show if the statistical test in Theorem 2 leads to the same conclusion as the bootstrapped 95% confidence interval regarding the existence of heteroscedastic noise?

Why should nCor_Delta be insensitive to the value of R^2? Is this a desirable property? Suppose R^2 becomes extremely close to 1, i.e. the noise level is much smaller than Var(f(x)), how would nCor_Delta behave?

One explanation for the worse performance of nCor_Delta when X becomes multivariate could be the curse of dimensionality makes it more difficult to identify the nearest neighbors with extremely close Euclidean distance. Since the number of samples is also finite in real-world applications, would it be possible to establish a bound of sample complexity, i.e., the minimal number of samples required to reliably test the existence of the heteroscedastic noise, even in the one-dimensional case?

**Time Spent Reviewing:**

2

---

> ### Author Response · Authors · 2021-08-10
> **Response to the Official Review by Reviewer Xjiq**
>
> Dear Reviewer Xjiq
>
> First of all. we’d like to thank you for your careful reading and valuable comments. Then we address major concerns below.
>
> Question: Could the author elaborate on how to combine nCor_Delta and nCor to better discover relationships between variables? For example, if nCor_Delta can detect the heteroscedastic noise between input X and output Y. Given this discovery, how to apply nCor to find a better functional relationship compared to directly applying nCor?
>
> Answer: Interaction effect is a kind of multivariate functional dependence which cannot be directly measured by any pairwise association measure. When dealing with high-dimensional datasets, discovering interactions is always more difficult and computational costly. In such a situation, we can firstly use the bivariate $nCor_{|\Delta|}({x_i},y)$ test to examine the heteroscedasticity of each pair of $(x_i,y)$ for screening the potential interacting $x_i$, and subsequently conduct the multivariate $nCor$ and the COI tests introduced in reference [28] to identify the exact interactions on the reduced variable subset that only contains the potential interacting $x_i$. This procedure will be obviously much more efficient than identifying interactions through detecting the full dataset. To explain how to combine $nCor_{|\Delta|}$ and $nCor$ tests to better discover relationships, we will add one more Remark (Remark 3) into the main manuscript.
>
> Question: In section 4.1, 50 datasets are created for each functional relationship. However, in real-world data, only a single realization is available. Can the authors show if the statistical test in Theorem 2 leads to the same conclusion as the bootstrapped 95% confidence interval regarding the existence of heteroscedastic noise?
>
> Answer: Thank you for your suggestion. We will conduct more experiments on two real datasets (a banknote authentication dataset and a concrete compressive strength dataset). The dependences between variables in these datasets are highly nonlinear, and include both heteroscedastic and homoscedastic relationships (which have been verified by both $nCor$ tests and regression analysis). We have used the bootstrap percentile method to verify the conclusions obtained by using the statistical test given in Theorem 2. The two sets of conclusions are entirely the same. We will include these experiments details and results in supplementary material, and add the conclusions and discussions in the main manuscript.
>
> Question: Why should nCor_Delta be insensitive to the value of R^2? Is this a desirable property? Suppose R^2 becomes extremely close to 1, i.e. the noise level is much smaller than Var(f(x)), how would nCor_Delta behave?
>
> Answer: Yes, it is a desirable property, only then can $nCor_{|\Delta|}$ better discriminate between the nonfunctional and functional relationships. In addition, we have tested the performance of $nCor_{|\Delta|}$ with extremely large $R^2$ (0.9, 0.95, 0.97, 0.98, 0.99,0.999,0.9999). We found that when $R^2\geq 0.99$, the $nCor_{|\Delta|}$ test leads to more type 1 errors, that is, incorrectly reject the null hypothesis of homoscedastic. This is because when more than 99% variance of $y$ is predictable from $x$ by a functional relationship, the underlying residuals become too small to detected properly. Many thanks for your advice, we will clearly claim and discuss this issue in the main manuscript and add the details and results of the extra experiments in supplementary material.
>
> Question: One explanation for the worse performance of nCor_Delta when X becomes multivariate could be the curse of dimensionality makes it more difficult to identify the nearest neighbors with extremely close Euclidean distance. Since the number of samples is also finite in real-world applications, would it be possible to establish a bound of sample complexity, i.e., the minimal number of samples required to reliably test the existence of the heteroscedastic noise, even in the one-dimensional case?
>
> Answer: A lower bound of sample complexity will be helpful, however, so far we have no idea on how to derive it. As shown in Figure 3, generally say, several hundreds of data points should be sufficient for conduct the test, but we don’t how to prove it mathematically.

---

> > ### Comment · Reviewer_Xjiq · 2021-08-16
> > **Response to rebuttal**
> >
> > Thank the author for addressing my comments in detail. I have no further question and recommend the acceptance of this work.

---

### Official Review · Reviewer_cGGJ · 2021-07-18

**Rating:** 6
**Confidence:** 3

**Summary:**

This paper proposed a statistical measure named the absolute neighbour difference based neighbour correlation coefficient, to detect the associations between variables through examining the heteroscedasticity of the unpredictable variation of dependent variables. The method can somehow measure nonfunctional relationships. The method is simple and easy to implement that does not rely on explicitly estimating the regression residuals or the dependencies between variables so that it is not restrict to any kind of model assumption.

**Limitations And Societal Impact:**

Yes

**Main Review:**

Heteroscedasticity is a complicated scenario in which the noise level of measurement is controled by (or dependent on) a function g() of the independent variable, and therefore difficult to estimate. The authors examined carefully the variations of the  variable difference and proposed a new statistical test method todetect the heteroscedasticity relation. Overall this is an interesting work and the proposed method looks reasonable. I have a few detailed questions/suggestions

1. The authors claimed that heteroscedasticity can also arise from interacting variables, besides the multiplicative relation between the normalized nosie term Theta and the independent variable Xi. Then is it necessary to discriminate between these two scenarios or is it mathematically infeasible to do so? From practical point of view they are of different values.

2. The proposed method is still based on pairs of variable. In case of multiple (more than 2) interacting independent variables, can you apply it to obtain the high-order interaction?

3. I would suggest the authors clearly describe the algorithm by listing each step (with involved variables) one by one, in order to facilitate the understanding of the readers. Currently the procedures and formula appears throughout the paper and appears a bit distributed. An ``algorithm'' figure summarizing the procedures would be helpful.

**Time Spent Reviewing:**

3

---

> ### Author Response · Authors · 2021-08-10
> **Response to the Official Review by Reviewer cGGJ**
>
> Dear Reviewer cGGJ
>
> First of all. we’d like to thank you for your careful reading and valuable comments. Then we address major concerns below.
>
> Question: 1- The authors claimed that heteroscedasticity can also arise from interacting variables, besides the multiplicative relation between the normalized nosie term Theta and the independent variable Xi. Then is it necessary to discriminate between these two scenarios or is it mathematically infeasible to do so? From practical point of view they are of different values.
>
> Answer: By only investigating the relationship between $(x_i,y)$, It is impossible to diagnose whether the heteroscedasticity arises from the multiplicative relation with unknown noise or with other interacting variables. $nCor_{|\Delta|}$ is unable to directly discriminate between these two scenarios, and an interaction can only be identified when all the interacting variables have been involved in the test.
>
> However, $nCor_{|\Delta|}$ still can provide useful information and indications. Consider two independent variables $x_i$ and $x_j$, if both $nCor_{|\Delta|}(x_i,y)$ and $nCor_{|\Delta|}(x_j,y)$ are significant, then an interaction probably exists. Subsequently, we can conduct further association tests to identify the exact interaction. For example, if $nCor(x_ix_j,y)$ is significant and much greater than $nCor(x_i,y)+nCor(x_j,y)$, then there is an interaction effect existing between $ (x_i,x_j)$ and $y$;  if $nCor_{|\Delta|} (x_ix_j,y)$ is insignificant, then the heteroscedasticity only arises from the interaction of $ (x_i,x_j)$; or, if $nCor_{|\Delta|} (x_ix_j,y)$ is still significant, then a higher-order interaction may occurs.
>
> Question: 2- The proposed method is still based on pairs of variable. In case of multiple (more than 2) interacting independent variables, can you apply it to obtain the high-order interaction?
>
> Answer: The proposed method can be used for detecting heteroscedastic relationship with multiple independent variables. A significant $nCor_{|\Delta|}({\bf x},y)$ where $\||{\bf x}\||>1$ implies that an interaction of at least $\||{\bf x}\||+1$ independent variables probably exists. However, it still cannot identify the high-order interaction directly. For high-dimensional dataset, we suggest to use the bivariate $nCor_{|\Delta|}({x_i},y)$ test firstly to examine the heteroscedasticity of each pairwise relationship $(x_i,y)$ for screening the potential interacting $x_i$, and then conduct thorough $nCor$ test to identify the exact interactions on the reduced variable subset that only contains the potential interacting $x_i$. This procedure may significantly reduce the computational effort for identifying interactions. We will add one more Remark (Remark 3) into the main manuscript for explaning and discussing this issue.
>
> Question: 3-I would suggest the authors clearly describe the algorithm by listing each step (with involved variables) one by one, in order to facilitate the understanding of the readers. Currently the procedures and formula appears throughout the paper and appears a bit distributed. An ``algorithm'' figure summarizing the procedures would be helpful.
>
> Answer: Thank you for your suggestion, we will add a pseudo code (summarizing the computational procedure of the proposed method) into the main manuscript to make it more clear.

---

> > ### Comment · Reviewer_cGGJ · 2021-08-17
> > **Thanks for your response**
> >
> > I have no further questions about the paper.

---

### Official Review · Reviewer_v9y1 · 2021-07-20

**Rating:** 7
**Confidence:** 4

**Summary:**

This paper proposes a correlation test to detect if heteroscedasticity exists between X and Y.   The method makes use of the neighboring variables and take the pearson correlation between $\Delta$ Y and $\Delta$ Y".   This test does not require the computation of the regression residuals that requires well-specified model assumptions.  The authors claim that this test was able to successfully detected the heteroscedasticity of the data.

**Limitations And Societal Impact:**

It would be good if the authors can include some discussions about the potential applications of the detection heteroscedastic relationships.

**Main Review:**

This paper presents an interesting idea to detect any heteroscedastic relationships between variables.  The principle behind is very simple but it is believed to be novel.  This work may have significant impact if there are practical examples that require the detection of heteroscedastic relationships.  The key idea is that the neighboring data point assumption enables us to focus on the residuals directly without the need of finding the regression residuals,.  The test is developed base on the relationship described in equation 1 which is a general model form.   One key point is that all data points have to be close to each other, otherwise the assumption does not hold.  Therefore, it is not surprising that it works well in single dimension case when data are relatively close to each other.  However, as the dimension increases, due to the curse of dimensionality, we need a much larger dataset in order to guarantee the closest property.   Besides, in high dimensional cases, it is said that the optimal reordering permutation is obtained by minimizing the total distance between each pair of neighboring data points and nearest neighbor algorithm is used to obtain the optimal or sub-optimal permutation.  On one hand, it is quite demanding to compute the permutation for large datasets.  On the other hand, the sub-optimal results may affect the closest assumption as well.

The paper is nicely written and well organized.  The experiments are clearly presented and evaluated.   It is expected that the results can easily be reproduced.   In the experiment, $f(x) = \beta \sin(\pi  x) $ and $g(x)$ is related to $ \cos(\pi x) $ in examples 3 and 4.  f(x) and g(x) have the same frequency.  I wonder if it is more difficult to detect the heteroscedastic relationships if  f(x) and g(x) have different frequencies, say $g(x)$ is related to $ \cos(n \pi x) $ instead.

In line 118, it says Lemma 2.1.  Should it be Lemma 1 ?

**Time Spent Reviewing:**

6 hours

---

> ### Author Response · Authors · 2021-08-10
> **Response to the Official Review by Reviewer v9y1**
>
> Dear Reviewer v9y1
>
> First of all. we’d like to thank you for your careful reading and valuable comments. Then we address major concerns below.
>
> Question: One key point is that all data points have to be close to each other, otherwise the assumption does not hold. On one hand, it is quite demanding to compute the permutation for large datasets. On the other hand, the sub-optimal results may affect the closest assumption as well.
>
> Answer: We agree. The key point is that the paired data points obtained by data reordering have to be close to each other. The reordering permutation and its total length may impact on the value of $nCor_{|\Delta|}$. Fortunately, NN algorithm can always find a sub-optimal permutation that is of sufficient quality for conducting $nCor_{|\Delta|}$ detection, that is, the distances between the overwhelming majority of the neighboring data points reordered by the permutation should be sufficiently small. Actually, we did some experiments before to check it, and the results showed that the $nCor_{|\Delta|}$ test is almost robust to the different permutations obtained by changing the initial condition of NN algorithm.
>
> Question: In the experiment, f(x)=βsin⁡(πx) and g(x) is related to cos⁡(πx) in examples 3 and 4. f(x) and g(x) have the same frequency. I wonder if it is more difficult to detect the heteroscedastic relationships if f(x) and g(x) have different frequencies, say g(x) is related to cos⁡(nπx) instead.
>
> Answer: The detection power of $nCor_{|\Delta|}$ doesn’t rely on the frequencies of f(x) and g(x), and the proposed method can still detect the relationships even though the $sin(\cdot)$ and $cos(\cdot)$ functions have different frequencies.
>
> In line 118, it says Lemma 2.1. Should it be Lemma 1 ?
>
> Answer: We apologize for the typo, and it will be corrected in the final version.
>
> Question: It would be good if the authors can include some discussions about the potential applications of the detection heteroscedastic relationships.
>
> Answer: Thank you for your suggestion, we will add more discussions (in Section 5) on how to use the proposed measure to assist in both feature selection and model validation.

---

### Official Review · Reviewer_53UA · 2021-07-20

**Rating:** 6
**Confidence:** 2

**Summary:**

The work studies the detection of "heteroscedastic relationships" in data samples. Based on the previous model of "neighbour correlation coefficien (nCor)", the work further proposes an "absolute neighbour difference based neighbour correlation coefficient (nCor_|△|)" and shows some properties of the coefficient, supported with preliminary experimental results on artificial data samples.

**Limitations And Societal Impact:**

See "Main Review"

**Main Review:**

Pros:

--1. I am not familiar with the problem of "heteroscedasticity" that is studied in the paper, while it seems a meaningful problem to me.

--2. The writing of the paper is not difficult to follow, although there is much room to improve.

Concerns:

--1. This work seems a marginal improvement to Zhang's paper at AAAI'2020 (ie reference [28]). The key result of "nCor" test has been established there. The new proposal of "nCor_|△|" seems not that significant to me. Correct me if I ignored important issues or further justifications on this point are expected in the rebuttal phase.

--2. It seems to me that the proposed method might need a lot of data samples. Further justification or illustration on this point will be beneficial to readers to fully understand the proposed method.

--3. In addition to artificial datasets, evaluation results on high-dimensional real datasets are strongly encouraged. Furthermore, it would be desirable to see if the author(s) could give some examples that the proposed model succeed but the previous testing methods fail to detect.

In line 73, the citation/reference to the key definition of "heteroscedasticity" seems to be problemastic.

==================
The author(s) addressed my concerns in the rebuttal and I tend to accept the work.

**Time Spent Reviewing:**

~5

---

> ### Author Response · Authors · 2021-08-10
> **Respond to the Official Review by Reviewer 53UA**
>
> Dear Reviewer 53UA
>
> First of all. we’d like to thank you for your careful reading and valuable comments. Then we address major concerns below.
>
> Question: --1. This work seems a marginal improvement to Zhang's paper at AAAI'2020 (ie reference [28]). The key result of "nCor" test has been established there. The new proposal of "nCor_|△|" seems not that significant to me. Correct me if I ignored important issues or further justifications on this point are expected in the rebuttal phase.
>
> Answer: $nCor_{|\Delta|}$ is a novel heteroscedasticity test and focus on detecting the nonfunctional relationships between variables only. So that it is quite different from $nCor$ (which can only detect functional relationships) and other previous approaches such as MI and MIC (fail to discriminate between the two types of relationships). Although there is no functional relationship ($nCor$ is insignificant), heteroscedastic relationship (nonfunctional relationship) may exist and $nCor_{|\Delta|}$ is still able to detect the dependence in such a situation. Therefore, the two methods are not alternatives to each other, but rather use for different purposes.
>
> Question: --2. It seems to me that the proposed method might need a lot of data samples. Further justification or illustration on this point will be beneficial to readers to fully understand the proposed method.
>
> Answer: Yes, it need a comparatively large sample size. As shown in Figure 3, in general, several hundreds of data points should be sufficient for conduct the test. We will further clarify this issue in Section 4,1 in the main manuscript. In addition, for further illustratre the impact of sample size on the detection power of the proposed method, we will extend the experiments in Section 4.1 with three smaller sample sizes (30, 50, and 100), and add the extra experimental results into supplementary material.
>
> Question: --3. In addition to artificial datasets, evaluation results on high-dimensional real datasets are strongly encouraged. Furthermore, it would be desirable to see if the author(s) could give some examples that the proposed model succeed but the previous testing methods fail to detect.
>
> Answer: In supplementary material there was a simple example showed that the proposed method could easily discover interesting heteroscedastic relationships and interactions in a high-dimensional real dataset. Moreover, we will perform more experiments on two real datasets (banknote authentication data and concrete compressive strength data), and we will add the experiment details and results in supplementary material to further evaluate the performance of the new method. Of course, except the regression modelling based heteroscedasticity tests, none of the existing methods can clearly detect or discriminate these nonfunctional relationships from the functional ones.
>
> Question: In line 73, the citation/reference to the key definition of "heteroscedasticity" seems to be problemastic.
>
> Answer: We apologize for the typo, and it will be corrected in the final version.

---

> > ### Comment · Reviewer_53UA · 2021-08-17
> > **response to rebuttal**
> >
> > Thanks for the reply. The clarification addressed my concerns. I have no further questions and tend to accept the work now.

---

### Official Review · Reviewer_sHbg · 2021-08-02

**Rating:** 6
**Confidence:** 3

**Summary:**

This paper proposes a new test for heteroscedasticity.  Heteroscedasticity is an important issue in real-world applications that in my opinion is under-considered and under-addressed.  The paper presents theoretical results on type I and type II error and provides significant empirical results for the benefits of the method over others such as the Park test and the White test.  It discusses limitations such as decreased relative power in high-dimensional settings.

**Limitations And Societal Impact:**

Yes, very good discussion of limitations.  More accurately understanding the relationships among variables can be important for social impact.

**Main Review:**

To my knowledge the proposed test is novel.  The empirical results and theoretical results show it is promising, especially in the motivating setting where we wish to know if one variable affects the spread of values for another variable.  The paper is intuitive and enjoyable to read, with a nice mix of theory and empirical results.  The paper also assesses its limitations.

One concern is that most real-world applications now are high-dimensional, and the methods seems to have weaknesses in this setting relative to other methods, as the paper accurately acknowledges and discusses.

**Time Spent Reviewing:**

one hour

---

> ### Author Response · Authors · 2021-08-10
> **Response to the Official Review by Reviewer sHbg**
>
> Dear Reviewer sHbg
>
> First of all. we’d like to thank you for your careful reading and valuable comments. Then we address major concern below.
>
> Question: One concern is that most real-world applications now are high-dimensional, and the methods seems to have weaknesses in this setting relative to other methods, as the paper accurately acknowledges and discusses.
>
> Answer: $nCor_{|\Delta|}({\bf x},y)$ displays much less detection power when $\||{\bf x}\||\gg 1$, since it is an association measure for detecting heteroscedastic relationships which are always more complicated than the functional ones.
>
> However, it is still helpful when dealing with high-dimensional datasets. Suppose we have a dataset that contains complex interactions (which are always more difficult to detect). In such a situation, we could firstly use bivariate $nCor_{|\Delta|}({x_i},y)$ test to examine the heteroscedasticity of each pairwise relationship $(x_i,y)$ to find all the potential interacting $x_i$, and subsequently conduct thorough $nCor$ test to identify the exact interactions on the reduced variable subset that only includes the potential interacting $x_i$. This procedure may significantly reduce the computational cost for discovering the interactions, and help us  identify the multivariate relationships more efficiently.
> We will add more explanations and discussions on this issue (as Remark 3) into the main manuscript.

---

### Decision · Program_Chairs · 2021-09-27

**Decision:**

Accept (Poster)

**Comment:**

This paper proposes a measure called absolute neighbour difference-based neighbour correlation coefficient, to detect the "heteroscedasticity" in the nonfunctional relationship between variables. All reviewers agree that the detection of "heteroscedastic relations" is an important issue in learning and that the proposed method, although simple and intuitive, is novel and has practical implications.